# A Measure of Synergy Based on Union Information

**DOI:** 10.3390/e26030271

**Published:** 2024-03-19

**Authors:** André F. C. Gomes, Mário A. T. Figueiredo

**Affiliations:** Instituto de Telecomunicações and LUMLIS (Lisbon ELLIS Unit), Instituto Superior Técnico, Universidade de Lisboa, 1049-001 Lisboa, Portugal; mario.figueiredo@tecnico.ulisboa.pt

**Keywords:** information theory, partial information decomposition, union information, synergy, communication channels

## Abstract

The *partial information decomposition* (PID) framework is concerned with decomposing the information that a set of (two or more) random variables (the sources) has about another variable (the target) into three types of information: unique, redundant, and synergistic. Classical information theory alone does not provide a unique way to decompose information in this manner and additional assumptions have to be made. One often overlooked way to achieve this decomposition is using a so-called measure of union information—which quantifies the information that is present in at least one of the sources—from which a synergy measure stems. In this paper, we introduce a new measure of union information based on adopting a communication channel perspective, compare it with existing measures, and study some of its properties. We also include a comprehensive critical review of characterizations of union information and synergy measures that have been proposed in the literature.

## 1. Introduction

Williams and Beer [1] introduced the *partial information decomposition* (PID) framework as a way to characterize, or analyze, the information that a set of random variables (often called *sources*) has about another variable (referred to as the *target*). PID is a useful tool for gathering insights and analyzing the way information is stored, modified, and transmitted within complex systems [2,3]. It has been applied in several areas such as cryptography [4] and neuroscience [5,6], with many other potential use cases, such as in studying information flows in gene regulatory networks [7], neural coding [8], financial markets [9], and network design [10,11].

Consider the simplest case, a three-variable joint distribution $p(y_{1},y_{2},t)$ describing three random variables: two so-called sources, $Y_{1}$ and $Y_{2}$, and a target *T*. Notice that, despite what the names *sources* and *target* might suggest, there is no directionality (causal or otherwise) assumption. The goal of PID is to *decompose* the information that the sources $Y=(Y_{1},Y_{2})$ have about *T* into the sum of four non-negative quantities: the information that is present in both $Y_{1}$ and $Y_{2}$, known as *redundant* information, *R*; the information that only $Y_{1}$ (respectively $Y_{2}$) has about *T*, known as *unique* information, $U_{1}$ (respectively $U_{2}$); and the *synergistic* information, *S*, that is present in the pair $\left(Y_{1},Y_{2}\right)$ but not in $Y_{1}$ or $Y_{2}$ alone. In this case with two variables, the goal is, thus, to write
(1)$$I\left(T;Y\right)=R+U_{1}+U_{2}+S,$$
where I(T;Y) is the mutual information between *T* and *Y* [12]. The redundant information *R*, because it is present in both $Y_{1}$ and $Y_{2}$, is also referred to as *intersection* information and denoted as $I_{\cap }$. Finally, $I_{\cup }$ refers to *union* information, i.e., the amount of information provided by at least one of the sources; in the case of two sources, $I_{\cup }=U_{1}+U_{2}+R$, thus $S=I\left(T;Y\right)-I_{\cup }$.

Because unique information and redundancy satisfy the relationship $U_{i}=I\left(T;Y_{i}\right)-R$ (for i∈{1,2}), it turns out that defining how to compute one of these quantities (*R*, $U_{i}$, or *S*) is enough to fully determine the others [1]. Williams and Beer [1] suggested a set of axioms that a measure of redundancy should satisfy, and proposed a measure of their own. Those axioms became well known as the Williams–Beer axioms, although the measure they proposed has subsequently been criticized for not capturing informational content, but only information *size* [13]. It is worth noting that as the number of variables grows the number of terms appearing in the PID of I(T;Y) grows super-exponentially [14].

Stimulated by that initial work, other measures of information and other sets of axioms for information decomposition have been introduced; see, for example, the work by Bertschinger et al. [15], Griffith and Koch [16], and James et al. [17], for different measures of redundant, unique, and synergistic information. There is no consensus about what axioms any measure should satisfy or whether a given measure *captures the information* that it should capture, except for the Williams–Beer axioms. Today, there is still debate about what axioms different measures of information should satisfy, and there is no general agreement on what is an appropriate PID [17,18,19,20,21].

Most PID measures that have been suggested thus far are either measures of redundant information, e.g., [1,13,21,22,23,24,25,26], or measures of unique information, e.g., [15,17]. Alternatively, it is possible to define the *union information* of a set of sources as the amount of information provided by at least one of those sources. Synergy is then defined as the difference between the total information and union information [22].

In this paper, we introduce a new measure of union information based on the information channel perspective that we already pursued in earlier work [26] and study some of its properties. The resulting measure leads to a novel information decomposition that is particularly suited for analyzing how information is distributed in channels.

The rest of the paper is organized as follows. A final subsection of this section introduces the notation used throughout the paper. In Section 2, we recall some properties of PID and take a look at how the degradation measure for redundant information introduced by Kolchinsky [22] decomposes information in bivariate systems, while also pointing out some drawbacks of that measure. Section 3 presents the motivation for our proposed measure, its operational interpretation, its multivariate definition, as well as some of its drawbacks. In Section 4, we propose an extension of the Williams–Beer axioms for measures of union information and show that our proposed measure satisfies those axioms. We review all properties that have been proposed both for measures of union information and synergy, and either accept or reject them. We also compare different measures of synergy and relate them, whenever possible. Finally, Section 5 presents concluding remarks and suggestions for future work.

### Notation

For two discrete random variables X∈X and Z∈Z, their Shannon mutual information I(X;Z) is given by I(X;Z)=I(Z;X)=H(X)−H(X|Z)=H(Z)−H(Z|X), where $H\left(X\right)=-\sum_{x\in X}p\left(x\right)\log p\left(x\right)$ and $H\left(X|Z\right)=-\sum_{x\in X}\sum_{z\in X}p\left(x,z\right)\log p\left(x|z\right)$ are the entropy and conditional entropy, respectively [12]. The conditional distribution p(z|x) corresponds, in an information-theoretical perspective, to a discrete memoryless channel with a channel matrix *K*, i.e., such that K[x,z]=p(z|x) [12]. This matrix is row-stochastic: K[x,z]≥0, for any x∈X and z∈Z, and $\sum_{z\in Z}K\left[x,z\right]=1$, for any *x*.

Given a set of *n* discrete random variables (sources), $Y_{1}\in Y_{1},\ldots ,Y_{n}\in Y_{n}$, and a discrete random variable T∈T (target) with joint distribution (probability mass function) $p(y_{1},\ldots ,y_{n},t)$, we consider the channels $K^{\left(i\right)}$ between *T* and each $Y_{i}$, that is, each $K^{\left(i\right)}$ is a $\left|T|\times \right|Y_{i}|$ row-stochastic matrix with the conditional distribution $p(y_{i}|t)$. For a vector *y*, $y_{-i}$ refers to the same vector without the $i^{th}$ component.

We say that three random variables, say X,Y,Z, form a Markov chain (which we denote by X−Y−Z or by X⊥Z∣Y) if *X* and *Z* are conditionally independent, given *Y*.

## 2. Background

### 2.1. PID Based of Channel Orders

In its current versions, PID is agnostic to causality in the sense that, like mutual information, it is an undirected measure, i.e., I(T;Y)=I(Y;T). Some measures indirectly presuppose some kind of directionality to perform PID. Take, for instance, the redundancy measure introduced by Kolchinsky [22], based on the so-called degradation order ${⪯}_{d}$ between communication channels (see recent work by Kolchinsky [22] and Gomes and Figueiredo [26] for definitions):(2)$$I_{\cap }^{d}\left(Y_{1},Y_{2},\ldots ,Y_{m}\to T\right):=\underset{K^{Q}:\;K^{Q}{⪯}_{d}K^{\left(i\right)},\;i\in \left\{1,\ldots ,m\right\}}{\sup}I\left(Q;T\right).$$
In the above equation, *Q* is the output of the channel T→Q, which we also denote as $K^{Q}$. When computing the information shared by the *m* sources, $I_{\cap }^{d}\left(Y_{1},Y_{2},\ldots ,Y_{m}\to T\right)$, the perspective is that there is a channel with a single input *T* and *m* outputs $Y_{1},\ldots ,Y_{m}$. This definition of $I_{\cap }^{d}$ corresponds to the mutual information of the most informative channel, under the constraint that this channel is dominated (in the degradation order ${⪯}_{d}$ sense) by all channels $K^{\left(1\right)},\ldots ,K^{\left(m\right)}$. Since mutual information was originally introduced to formalize the capacity of communication channels, it is not surprising that measures that presuppose channel directionality are found useful in this context. For instance, the work of James et al. [27] concluded that only if one assumes the directionality $T\to Y_{i}$ does one obtain a valid PID from a secret key agreement rate, which supports the approach of assuming this directionality perspective.

Although it is not guaranteed that the structure of the joint distribution $p(y_{1},\ldots ,y_{n},t)$ is compatible with the causal model of a single input and multiple output channels (which implies that the sources are conditionally independent, given *T*), one may always compute such measures, which have interesting and relevant operational interpretations. In the context of PID, where the goal is to study how information is decomposed, such measures provide an excellent starting point. Although it is not guaranteed that there is actually a channel (or a direction) from *T* to $Y_{i}$, we can characterize how information about *T* is spread through the sources. In the case of the degradation order, $I_{\cap }^{d}$ provides insight about the maximum information obtained about *T* if any $Y_{i}$ is observed.

Arguably, the most common scenario in PID is finding out something about the structure of the information the variables $Y_{1},\ldots ,Y_{n}$ have about *T*. In a particular system of variables characterized by its joint distribution, we do not make causal assumptions, so we can adopt the perspective that the variables $Y_{i}$ are functions of *T*, hence obtaining the channel structure. Although this channel structure may not be *faithful* [28] to the conditional independence properties implied by $p(y_{1},\ldots ,y_{n},t)$, this channel perspective allows for decomposition of I(Y;T) and for drawing conclusions about the inner structure of the information that *Y* has about *T*. Some distributions, however, *cannot* have this causal structure. Take, for instance, the distribution generated by $T=Y_{1}\;\mathrm{xor}\;Y_{2}$, where $Y_{1}$ and $Y_{2}$ are two equiprobable and independent binary random variables. We will call this distribution XOR. For this well-known distribution, we have $Y_{1}⊥Y_{2}$ and Y1⊥Y2|T, whereas the implied channel distribution that $I_{\cap }^{d}$ assumes yields the exact opposite dependencies, that is, Y1⊥Y2 and $Y_{1}⊥Y_{2}|T$; see Figure 1 for more insight.

Consider the computation of $I_{\cap }^{d}\left(Y_{1},Y_{2}\to T\right)$ for the XOR distribution. This measure argues that, since
$$K^{\left(1\right)}=K^{\left(2\right)}=\left[\begin{array}{cc} 0.5 & 0.5\\ 0.5 & 0.5 \end{array}\right],$$
then a solution to $I_{\cap }^{d}\left(Y_{1},Y_{2}\to T\right)$ is given by the channel $K^{Q}=K^{\left(1\right)}$ and redundancy is computed as I(Q;T), yielding 0 bits of redundancy, and consequently 1 bit of synergy (as computed from (Equation 1)). Under this channel perspective (as in Figure 1b), $I_{\cap }^{d}$ is not concerned with, for example, $p(Y_{1}|T,Y_{2})$ or $p(Y_{1},Y_{2})$. If all that is needed to compute redundancy is $p\left(T\right),K^{\left(1\right)}$, and $K^{\left(2\right)}$, this would lead to the wrong conclusion that the outcome $\left(Y_{1},Y_{2},T\right)=\left(0,0,1\right)$ has non-null probability, which it does not. With this, we do not mean that $I_{\cap }^{d}$ is an incomplete or incorrect measure to perform PID, we are using its insights to point us in a different direction.

### 2.2. Motivation for a New PID

At this point, the most often used approaches to PID are based on redundancy measures. Usually, these are in one of the two following classes:Measures that are not concerned with information content, only *information size*, which makes them easy to compute even for distributions with many variables, but at the cost that the resulting decompositions may not give realistic insights into the system, precisely because they are not sensitive to informational content. Examples are $I_{\cap }^{\mathrm{WB}}$ [1] or $I_{\cap }^{\mathrm{MMI}}$ [23], and applications of these can be found in [29,30,31,32].Measures that satisfy the Blackwell property—which arguably do measure information content—but are insensitive to changes in the sources’ distributions $p\left(y_{1},\ldots ,y_{n}\right)=\sum_{t}p\left(y_{1},\ldots ,y_{n},t\right)$ (as long as $p\left(T\right),K^{\left(1\right)},\ldots ,K^{\left(n\right)}$ remain the same). Examples are $I_{\cap }^{d}$ [22] (see Equation (Equation 2)) or $I_{\cap }^{\mathrm{BROJA}}$ [15]. It should be noted that $I_{\cap }^{\mathrm{BROJA}}$ is only defined for the bivariate case, that is, for distributions with at most two sources, described by $p(y_{1},y_{2},t)$. Applications of these can be found in [33,34,35].

Particularly, $I_{\cap }^{d}$ and $I_{\cap }^{\mathrm{BROJA}}$ satisfy the so-called (*) assumption [15], which argues that redundant and unique information should only depend on the marginal distribution of the target p(T) and on the conditional distributions of the sources given the target, that is, on the stochastic matrices $K^{\left(i\right)}$. James et al. [17] (Section 4) and Ince [21] (Section 5) provide great arguments as to why the (*) assumption should not hold in general, and we agree with them.

Towards motivating a new PID, let us look at how $I_{\cap }^{d}$ decomposes information in the bivariate case. Any measure that is based on a preorder between channels and which satisfies Kolchinsky’s axioms yields similar decompositions [26], thus there is no loss of generality in focusing on $I_{\cap }^{d}$. We next analyze three different cases.

Case 1: There is an ordering between the channels, that is, w.l.o.g., $K^{\left(2\right)}{⪯}_{d}K^{\left(1\right)}$. This means that $I\left(Y_{2};T\right)\leq I\left(Y_{1};T\right)$ and the decomposition (as in (Equation 1)) is given by $R=I(Y_{2};T)$, $U_{2}=0$, $U_{1}=I\left(Y_{1};T\right)-I\left(Y_{2};T\right)$, and $S=I\left(Y;T\right)-I\left(Y_{1};T\right)$. Moreover, if K(1)⪯dK(2), then S=0.As an example, consider the leftmost distribution in Table 1, which satisfies $T=Y_{1}$. In this case,
$$K^{\left(1\right)}=\left[\begin{array}{cc} 1 & 0\\ 0 & 1 \end{array}\right]{⪰}_{d}K^{\left(2\right)}=\left[\begin{array}{cc} 0.5 & 0.5\\ 0.5 & 0.5 \end{array}\right],$$
yielding R=0, $U_{1}=1$, $U_{2}=0$, and S=0, as expected, because $T=Y_{1}$.

**Table 1 entropy-26-00271-t001:** Three joint distributions $p(yt,y_{2},y_{2})$ used to exemplify the three cases. Left: joint distribution satisfying $T=Y_{1}$. Middle: distribution satisfying $T=(Y_{1},Y_{2})$, known as the COPY distribution. Right: the so-called BOOM distribution (see text).

*t*	$y_{1}$	$y_{2}$	$p(t,y_{1},y_{2})$	*t*	$y_{1}$	$y_{2}$	$p(t,y_{1},y_{2})$	*t*	$y_{1}$	$y_{2}$	$p(t,y_{1},y_{2})$
0	0	0	1/4	(0,0)	0	0	1/4	0	0	2	1/6
0	0	1	1/4	(0,1)	0	1	1/4	1	0	0	1/6
1	1	0	1/4	(1,0)	1	0	1/4	1	1	2	1/6
1	1	1	1/4	(1,1)	1	1	1/4	2	0	0	1/6
								2	2	0	1/6
								2	2	1	1/6

Case 2: There is no ordering between the channels and the solution of $I_{\cap }^{d}\left(Y_{1},Y_{2}\to T\right)$ is a trivial channel, in the sense that it has no information about *T*. The decomposition is given by R=0, $U_{2}=I\left(Y_{2};T\right)$, $U_{1}=I\left(Y_{1};T\right)$, and $S=I\left(Y;T\right)-I\left(Y_{1};T\right)-I\left(Y_{2};T\right)$, which may lead to a negative value of synergy. An example of this is provided later.As an example, consider the COPY distribution with $Y_{1}$ and $Y_{2}$ i.i.d. Bernoulli variables with parameter 0.5, shown in the center of Table 1. In this case, channels $K^{\left(1\right)}$ and $K^{\left(2\right)}$ have the forms
$$K^{\left(1\right)}=\left[\begin{array}{cc} 1 & 0\\ 1 & 0\\ 0 & 1\\ 0 & 1 \end{array}\right],\;K^{\left(2\right)}=\left[\begin{array}{cc} 1 & 0\\ 0 & 1\\ 1 & 0\\ 0 & 1 \end{array}\right],$$
with no degradation order between them. This yields R=0, $U_{1}=U_{2}=1$, and S=0.Case 3: There is no ordering between the channels and $I_{\cap }^{d}\left(Y_{1},Y_{2}\to T\right)$ is achieved by a nontrivial channel $K^{Q}$. The decomposition is given by R=I(Q;T), $U_{2}=I\left(Y_{2};T\right)-I\left(Q;T\right)$, $U_{1}=I\left(Y_{1};T\right)-I\left(Q;T\right)$, and $S=I\left(Y;T\right)+I\left(Q;T\right)-I\left(Y_{1};T\right)-I\left(Y_{2};T\right)$.As an example, consider the BOOM distribution [17], shown on the right-hand side of Table 1. In this case, channels $K^{\left(1\right)}$ and $K^{\left(2\right)}$ are
$$K^{\left(1\right)}=\left[\begin{array}{ccc} 1 & 0 & 0\\ 1/2 & 1/2 & 0\\ 1/3 & 0 & 2/3 \end{array}\right],\;K^{\left(2\right)}=\left[\begin{array}{ccc} 0 & 0 & 1\\ 1/2 & 0 & 1/2\\ 2/3 & 1/3 & 0 \end{array}\right],$$
and there is no degradation order between them. However, there is a nontrivial channel $K^{Q}$ that is dominated by both $K^{\left(1\right)}$ and $K^{\left(2\right)}$ that maximizes I(Q;T). One of its versions is
$$K^{Q}=\left[\begin{array}{ccc} 0 & 1 & 0\\ 0 & 3/4 & 1/4\\ 1/3 & 1/3 & 1/3 \end{array}\right],$$
yielding R≈0.322, $U_{1}=U_{2}\approx 0.345$, and S≈0.114.

This class of approaches has some limitations, as is the case for all PID measures. In the bivariate case, the definition of synergy *S* from a measure of redundant information is the completing term such that $I\left(Y;T\right)=S+R+U_{1}+U_{2}$ holds. The definition of $I_{\cap }^{d}$ supports the argument that if $K^{\left(2\right)}{⪯}_{d}K^{\left(1\right)}$ and K(1)⪯dK(2), then there is no synergy. This makes intuitive sense because, in this case, $T-Y_{1}-Y_{2}$ is a Markov chain (see Section 1 for the definition), consequently, $I\left(Y;T\right)=I\left(Y_{1};T\right)$, that is, $Y_{1}$ has the same information about *T* as the pair $Y=(Y_{1},Y_{2})$.

If there is no ${⪯}_{d}$ ordering between the channels, as in the COPY distribution (Table 1, middle), the situation is more complicated. We saw that the decomposition for this distribution yields R=0, $U_{1}=U_{2}=1$, and S=0. However, suppose we change the distribution such that T=(1,1) has probability 0 and the other outcomes have probability 1/3. For example, consider the distribution in Table 2. For this distribution, we have I(Y;T)≈1.585. Intuitively, we would expect that I(Y;T) would be decomposed as R=0, $U_{1}=U_{2}=I\left(Y;T\right)/2$, and S=0, just as before, so that the proportions $U_{i}/I\left(Y;T\right)$, for i∈{1,2}, in both distributions remain the same, whereas redundancy and synergy would remain zero. That is, we do not expect that removing one of the outcomes while maintaining the remaining outcomes equiprobable would change the types of information in the system. However, if we perform this and compute the decomposition yielded by $I_{\cap }^{d}$, we obtain R=0, $U_{1}=U_{2}=0.918\neq I\left(Y;T\right)/2$, and S=−0.251, i.e., a negative synergy, arguably meaningless.

There are still many open questions in PID. One of those questions is: Should measures of redundant information be used to measure synergy, given that they compute it as the completing term in Equation (Equation 1). We agree that using a measure of redundant information to compute the synergy in this way may not be appropriate, especially because the *inclusion–exclusion principle* (IEP) should not necessarily hold in the context of PID; see [22] for comments on the IEP.

With these motivations, we propose a measure of union information for PID that shares with $I_{\cap }^{d}$ the implicit view of channels. However, unlike $I_{\cap }^{d}$ and $I_{\cap }^{\mathrm{BROJA}}$—which satisfy the (*) assumption, and thus, are not concerned with the conditional dependencies in $p(y_{i}|t,y_{-i})$—our measure defines synergy as the information that cannot be computed from $p(y_{i}|t)$, but can be computed from $p(y_{i}|t,y_{-i})$. That is, we propose that synergy be computed as the information that is not captured by assuming conditional independence of the sources, given the target.

## 3. A New Measure of Union Information

### 3.1. Motivation and Bivariate Definition

Consider a distribution $p(y_{1},y_{2},t)$ and suppose there are two agents, agent 1 and agent 2, whose goal is to reduce their uncertainty about *T* by observing $Y_{1}$ and $Y_{2}$, respectively. Suppose also that the agents know p(t), and that agent *i* has access to its channel distribution $p(y_{i}|t)$. Many PID measures make this same assumption, including $I_{\cap }^{d}$. When agent *i* works alone to reduce the uncertainty about *T*, since it has access to p(t) and $p(y_{i}|t)$, it also knows $p(y_{i})$ and $p(y_{i},t)$, which allows it to compute $I(Y_{i};T)$: the amount of uncertainty reduction about *T* achieved by observing $Y_{i}$.

Now, if the agents can work together, that is, if they have access to $Y=(Y_{1},Y_{2})$, then they can compute I(Y;T), because they have access to $p(y_{1},y_{2}|t)$ and p(t). On the other hand, if the agents are not able to work together (in the sense that they are not able to observe *Y* together, but only $Y_{1}$ and $Y_{2}$, separately) yet can communicate, then they can construct a different distribution *q* given by $q\left(y_{1},y_{2},t\right):=p\left(t\right)p\left(y_{1}|t\right)p\left(y_{2}|t\right)$, i.e., a distribution under which $Y_{1}$ and $Y_{2}$ are conditionally independent given *T*, but have the same marginal p(t) and the same individual conditionals $p(y_{1}|t)$ and $p(y_{2}|t)$.

The form of *q* in the previous paragraph should be contrasted with the following factorization of *p*, which entails no conditional independence assumption: $p\left(y_{1},y_{2},t\right)=p\left(t\right)p\left(y_{1}|t\right)p\left(y_{2}|t,y_{1}\right)$. In this sense, we would propose to define union information, for the bivariate case, as follows:(3)$$\begin{array}{cc} \; & I_{\cup }\left(Y_{1}\to T\right)=I_{q}\left(Y_{1};T\right)=I_{p}\left(Y_{1};T\right),\\ \; & I_{\cup }\left(Y_{2}\to T\right)=I_{q}\left(Y_{2};T\right)=I_{p}\left(Y_{2};T\right),\\ \; & I_{\cup }\left(Y_{1},Y_{2}\to T\right)=I_{q}\left(Y;T\right),\\ \; & I_{\cup }\left(\left(Y_{1},Y_{2}\right)\to T\right)=I_{p}\left(Y;T\right), \end{array}$$
where the subscript refers to the distribution under which the mutual information is computed. From this point forward, the absence of a subscript means that the computation is performed under the true distribution *p*. As we will see, this is not yet the final definition, for reasons to be addressed below.

Using the definition of synergy derived from a measure of union information [22], for the bivariate case we have
(4)$$\begin{array}{c} S\left(Y_{1},Y_{2}\to T\right):=I\left(Y;T\right)-I_{\cup }\left(Y_{1},Y_{2}\to T\right). \end{array}$$
Synergy is often posited as *the difference between the whole and the union of the parts*. For our measure of union information, the ‘union of the parts’ corresponds to the reduction in uncertainty about *T*—under *q*—that agents 1 and 2 can obtain by sharing their conditional distributions. Interestingly, there are cases where the union of the parts is better than the whole, in the sense that $I_{\cup }\left(Y_{1},Y_{2}\to T\right)>I\left(Y;T\right)$. An example of this is given by the *Adapted ReducedOR* distribution, originally introduced by Ince [21] and adapted by James et al. [17], which is shown in the left-hand side of Table 3, where r∈[0,1]. This distribution is such that $I_{q}\left(Y;T\right)$ does not depend on *r* ($I_{q}\left(Y;T\right)\approx 0.549$), since neither p(t) nor $p(y_{1}|t)$ and $p(y_{2}|t)$ depend on *r*; consequently, $q(t,y_{1},y_{2})$ also does not depend on *r*, as shown in the right-hand side of Table 3.

It can be easily shown that if r>0.5, then $I_{q}\left(Y;T\right)>I\left(Y;T\right)$, which implies that synergy, if defined as in (Equation 4), could be negative. How do we interpret the fact that there exist distributions such that $I_{q}\left(Y;T\right)>I\left(Y;T\right)$? This means that under distribution *q*, which assumes $Y_{1}$ and $Y_{2}$ are conditionally independent given *T*, $Y_{1}$ and $Y_{2}$ reduce the uncertainty about *T* more than in the original distribution. Arguably, the parts working independently and achieving better results than the whole should mean there is no synergy, as opposed to negative synergy.

The observations in the previous paragraphs motivate our definition of a new measure of union information as
(5)$$\begin{array}{c} I_{\cup }^{\mathrm{CI}}\left(Y_{1},Y_{2}\to T\right):=\min\left\{I\left(Y;T\right),I_{q}\left(Y;T\right)\right\}, \end{array}$$
with the superscript CI standing for *conditional independence*, yielding a non-negative synergy:(6)$$\begin{array}{c} S^{\mathrm{CI}}\left(Y_{1},Y_{2}\to T\right)=I\left(Y;T\right)-I_{\cup }^{\mathrm{CI}}\left(Y_{1},Y_{2}\to T\right)=\max\left\{0,I\left(Y;T\right)-I_{q}\left(Y;T\right)\right\}. \end{array}$$
Note that for the bivariate case we have zero synergy if $p(t,y_{2},y_{2})$ is such that $Y_{1}{⊥}_{p}Y_{2}|T$, that is, if the outputs are indeed conditionally independent given *T*. Moreover, $I_{\cup }^{\mathrm{CI}}$ satisfies the monotonicity axiom from the extension of the Williams–Beer axioms to measures of union information (to be mentioned in Section 4.1), which further supports this definition. For the bivariate source case, the decomposition of I(Y;T) derived from a measure of union information is given by
$$\left\{\begin{array}{c} I_{\cup }\left(Y_{1}\right)=I\left(Y_{1};T\right)=U_{1}+R\\ I_{\cup }\left(Y_{2}\right)=I\left(Y_{2};T\right)=U_{2}+R\\ I_{\cup }\left(Y_{1},Y_{2}\right)=U_{1}+U_{2}+R\\ I_{\cup }\left(Y_{12}\right)=I\left(Y_{12};T\right)=U_{1}+U_{2}+R+S \end{array}\right.\Leftrightarrow \left\{\begin{array}{c} S=I\left(Y;T\right)-I_{\cup }\left(Y_{1},Y_{2}\right)\\ U_{1}=I_{\cup }\left(Y_{1},Y_{2}\right)-I\left(Y_{2};T\right)\\ U_{2}=I_{\cup }\left(Y_{1},Y_{2}\right)-I\left(Y_{1};T\right)\\ R=I\left(Y_{1};T\right)-U_{1}=I\left(Y_{2};T\right)-U_{2} \end{array}\right..$$

### 3.2. Operational Interpretation

For the bivariate case, if $Y_{1}$ and $Y_{2}$ are conditionally independent given *T* (Figure 1b), then $p(y_{1}|t)$ and $p(y_{2}|t)$ (and p(t)) suffice to reconstruct the original joint distribution $p(y_{1},y_{2},t)$, which means the union of the parts is enough to reconstruct the whole, i.e., there is no synergy between $Y_{1}$ and $Y_{2}$. Conversely, a distribution generated by the DAG in Figure 1c does not satisfy conditional independence (given *T*), hence we expect positive synergy, as is the case for the XOR distribution, and indeed our measure yields 1 bit of synergy for this distribution. These two cases motivate the operational interpretation of our measure of synergy: it is the amount of information that is not captured by assuming conditional independence of the sources (given the target).

Recall, however, that some distributions are such that $I_{q}\left(Y;T\right)>I_{p}\left(Y;T\right)$, i.e., such that the union of the parts ‘outperforms’ the whole. What does this mean? It means that under *q*, $Y_{1}$ and $Y_{2}$ have more information about *T* than under *p*: the constructed distribution *q*, which drops the conditional dependence of $Y_{1}$ and $Y_{2}$ given *T*, reduces the uncertainty that *Y* has about *T* more than the original distribution *p*. In some cases, this may happen because the support of *q* is larger than that of *p*, which may lead to a reduction in uncertainty under *q* that cannot be achieved under *p*. In these cases, since we are decomposing $I_{p}\left(Y;T\right)$, we revert to saying that the union information that a set of variables has about *T* is equal to $I_{p}\left(Y;T\right)$, so that our measure satisfies the monotonicity axiom (later introduced in Definition 2). We will comment on this compromise between satisfying the monotonicity axiom and ignoring dependencies later.

### 3.3. General (Multivariate) Definition

To extend the proposed measure to an arbitrary number n≥2 of sources, we briefly recall the synergy lattice [18,36] and the union information semi-lattice [36]. For n=3, these two lattices are shown in Figure 2. For the sake of brevity, we will not address the construction of the lattices or the different orders between sources. We refer the reader to the work of Gutknecht et al. [36] for an excellent overview of the different lattices, the orders between sources, and the construction of different PID measures.

In the following, we use the term *source* to mean a subset of the variables $\left\{Y_{1},\ldots ,Y_{n}\right\}$, or a set of such subsets, we drop the curly brackets for clarity and refer to the different variables by their indices, as is common in most works on PID. The decomposition resulting from a measure of union information is not as direct to obtain as one obtained from a measure of redundant information, as the solution for the information atoms is not a Möbius inversion [14]. One must first construct the measure of synergy for source α by writing
(7)$$\begin{array}{c} S^{\mathrm{CI}}\left(\alpha \to T\right)=I\left(Y;T\right)-I_{\cup }^{\mathrm{CI}}\left(\alpha \to T\right), \end{array}$$
which is the generalization of (Equation 6) for an arbitrary *source* α. In the remainder of this paper, we will often omit “→T” from the notation (unless it is explicitly needed), with the understanding that the target variable is always referred to as *T*. Also for simplicity, in the following, we identify the different agents that have access to different distributions as the distributions they have access to.

It is fairly simple to extend the proposed measure to an arbitrary number of sources, as illustrated in the following two examples.

**Example 1.** 
*To compute $I_{\cup }^{CI}\left(\left(Y_{1},Y_{2}\right),Y_{3}\right)$, agent $\left(Y_{1},Y_{2}\right)$ knows $p(y_{1},y_{2}|t)$, thus it can also compute, by marginalization, $p(y_{1}|t)$ and $p(y_{2}|t)$. On the other hand, agent $Y_{3}$ only knows $p(y_{3}|t)$. Recall that both agents also have access to p(t). By sharing their conditionals, the agents can compute $q_{1}\left(y_{1},y_{2},y_{3},t\right):=p\left(t\right)p\left(y_{1},y_{2}|t\right)p\left(y_{3}|t\right)$, and also $q_{2}\left(y_{1},y_{2},y_{3},t\right):=p\left(t\right)p\left(y_{1}|t\right)p\left(y_{2}|t\right)p\left(y_{3}|t\right)$. After this, they may choose whichever distribution has the highest information about T, while still holding the view that any information gain larger than I(Y;T) must be disregarded. Consequently, we write*

$$I_{\cup }^{CI}\left(\left(Y_{1},Y_{2}\right),Y_{3}\right)=\min\left\{I\left(Y;T\right),\max\left\{I_{q_{1}}\left(Y;T\right),I_{q_{2}}\left(Y;T\right)\right\}\right\}.$$



**Example 2.** 
*Slightly more complicated is the computation of $I_{\cup }^{CI}\left(\left(Y_{1},Y_{2}\right),\left(Y_{1},Y_{3}\right),\left(Y_{2},Y_{3}\right)\right)$. In this case, the three agents may compute four different distributions, two of which are the same $q_{1}$ and $q_{2}$ defined in the previous paragraph, and the other two are $q_{3}\left(y_{1},y_{2},y_{3},t\right):=p\left(t\right)p\left(y_{1},y_{3}|t\right)p\left(y_{2}|t\right)$, and $q_{4}\left(y_{1},y_{2},y_{3},t\right):=p\left(t\right)p\left(y_{1}|t\right)p\left(y_{2},y_{3}|t\right)$.*


Given these insights, we propose the following measure of union information.

**Definition 1.** 
*Let $A_{1},\ldots ,A_{m}$ be an arbitrary collection of m≥1 sources (recall sources may be subsets of variables). Assume that no source is a subset of another source and no source is a deterministic function of other sources (if there is, remove it). We define*

$$I_{\cup }^{CI}\left(A_{1},\ldots ,A_{m}\to T\right)=\min\left\{I\left(A;T\right),\max_{q\in Q}I_{q}\left(A;T\right)\right\},$$

*where $A=\overset{m}{\underset{i=1}{⋃}}A_{i}$ and Q is the set of all different distributions that the m agents can construct by combining their conditional distributions and marginalizations thereof.*


For instance, in Example 1 above, $A=\left\{Y_{1},Y_{2}\right\}\cup \left\{Y_{3}\right\}=\left\{Y_{1},Y_{2},Y_{3}\right\}$; in Example 2, $A=\left\{Y_{1},Y_{2}\right\}\cup \left\{Y_{1},Y_{3}\right\}\cup \left\{Y_{2},Y_{3}\right\}=\left\{Y_{1},Y_{2},Y_{3}\right\}$. In Example 1, $Q=\{q_{1},q_{2}\}$, whereas in Example 2, $Q=\{q_{1},q_{2},q_{3},q_{4}\}$. We now justify the conditions in Definition 1 and the fact that they do not entail any loss of generality.

The condition that no source is a subset of another source (which also excludes the case where two sources are the same) implies no loss of generality: if one source is a subset of another, say $A_{i}\subseteq A_{j}$, then $A_{i}$ may be removed without affecting either *A* or Q, thus yielding the same value for $I_{\cup }^{\mathrm{CI}}$. The removal of source $A_{i}$ is also performed for measures of intersection information, but under the opposite condition: whenever $A_{j}\subseteq A_{i}$.The condition that no source is a deterministic function of other sources is slightly more nuanced. In our perspective, an intuitive and desired property of measures of both union and synergistic information is that their value should not change whenever one adds a source that is a deterministic function of sources that are already considered. We provide arguments in favor of this property in Section 4.2.1. This property may not be satisfied by computing $I_{\cup }^{\mathrm{CI}}$ without previously excluding such sources. For instance, consider $p(t,y_{1},y_{2},y_{3})$, where $Y_{1}$ and $Y_{2}$ are two i.i.d. random variables following a Bernoulli distribution with parameter 0.5, $Y_{3}=Y_{2}$ (that is, $Y_{3}$ is deterministic function of $Y_{2}$), and $T=Y_{1}\;\mathrm{AND}\;Y_{2}$. Computing $I_{\cup }^{\mathrm{CI}}\left(Y_{1},Y_{2},Y_{3}\right)$ without excluding $Y_{3}$ (or $Y_{2}$) yields $I_{\cup }^{\mathrm{CI}}\left(Y_{1},Y_{2},Y_{3}\right)=I_{q}\left(Y_{1},Y_{2},Y_{3};T\right)\approx 0.6810$ and $I_{\cup }^{\mathrm{CI}}\left(Y_{1},Y_{2}\right)=I_{q}\left(Y_{1},Y_{2};T\right)\approx 0.5409$. This issue is resolved by removing deterministic sources before computing $I_{\cup }^{\mathrm{CI}}$.

We conclude this section by commenting on the monotonicity of our measure. Suppose we wish to compute the union information of sources $\left\{\left(Y_{1},Y_{2}\right),Y_{3}\right\}$ and $\left\{Y_{1},Y_{2},Y_{3}\right\}$. PID theory demands that $I_{\cup }^{\mathrm{CI}}\left(\left(Y_{1},Y_{2}\right),Y_{3}\right)\geq I_{\cup }^{\mathrm{CI}}\left(Y_{1},Y_{2},Y_{3}\right)$ (monotonicity of union information). Recall our motivation for $I_{\cup }^{\mathrm{CI}}\left(\left(Y_{1},Y_{2}\right),Y_{3}\right)$: there are two agents, the first has access to $p(y_{1},y_{2}|t)$ and the second to $p(y_{3}|t)$. The two agents assume conditional independence of their variables and construct $q^{\prime }\left(y_{1},y_{2},y_{3},t\right)=p\left(t\right)p\left(y_{1},y_{2}|t\right)p\left(y_{3}|t\right)$. The story is similar for the computation of $I_{\cup }^{\mathrm{CI}}\left(Y_{1},Y_{2},Y_{3}\right)$, in which case we have three agents that construct $q^{\prime \prime }\left(y_{1},y_{2},y_{3},t\right)=p\left(t\right)p\left(y_{1}|t\right)p\left(y_{2}|t\right)p\left(y_{3}|t\right)$. Now, it may be the case that $I_{q^{\prime }}\left(Y;T\right)<I_{q^{\prime \prime }}\left(Y;T\right)$; considering only these two distributions would yield $I_{\cup }^{\mathrm{CI}}\left(\left(Y_{1},Y_{2}\right),Y_{3}\right)<I_{\cup }^{\mathrm{CI}}\left(Y_{1},Y_{2},Y_{3}\right)$, contradicting monotonicity for measures of union information. To overcome this issue, for the computation of $I_{\cup }^{\mathrm{CI}}\left(\left(Y_{1},Y_{2}\right),Y_{3}\right)$—and other sources in general—the agent that has access to $p(y_{1},y_{2}|t)$ must be allowed to disregard the conditional dependence of $Y_{1}$ and $Y_{2}$ on *T*, even if it holds in the original distribution *p*.

## 4. Properties of Measures of Union Information and Synergy

### 4.1. Extension of the Williams–Beer Axioms for Measures of Union Information

As Gutknecht et al. [36] rightfully notice, the so-called Williams–Beer axioms [1] can actually be derived from parthood distribution functions and the consistency equation [36]. Consequently, they are not really axioms but consequences of the PID framework. As far as we know, there has been no proposal in the literature for the equivalent of the Williams–Beer axioms (which refer to measures of redundant information) for measures of union information. In the following, we extend the Williams–Beer axioms to measures of union information and show that the proposed $I_{\cup }^{\mathrm{CI}}$ satisfies these axioms. Although we just argued against calling them axioms, we keep the designation *Williams–Beer axioms* because of its popularity. Although the following definition is not in the formulation of Gutknecht et al. [14], we suggest formally defining union information from the formulation of parthood functions as
(8)$$I_{\cup }\left(Y_{1},\ldots ,Y_{m};T\right)=\sum_{\exists i:f(Y_{i})=1}\Pi \left(f\right),$$
where *f* refers to a parthood function and Π(f) is the information atom associated with the parthood function *f*. Given this formulation, the following properties must hold.

**Definition 2.** 
*Let $A_{1},\ldots ,A_{m}$ be an arbitrary number m≥2 of sources. A measure of union information $I_{\cup }$ is said to satisfy the Williams–Beer axioms for union information measures if it satisfies:*
*1.* 
*Symmetry: $I_{\cup }$ is symmetric in the $A_{i}$’s.*
*2.* 
*Self-redundancy: $I_{\cup }\left(A_{i}\right)=I\left(A_{i};T\right)$.*
*3.* 
*Monotonicity: $I_{\cup }\left(A_{1},\ldots ,A_{m-1},A_{m}\right)\geq I_{\cup }\left(A_{1},\ldots ,A_{m-1}\right)$.*
*4.* 
*Equality for monotonicity: $A_{m}\subseteq A_{m-1}\;\Rightarrow \;I_{\cup }\left(A_{1},\ldots ,A_{m-1},A_{m}\right)=I_{\cup }\left(A_{1},\ldots ,A_{m-1}\right)$.*



**Theorem 1.** 
*$I_{\cup }^{CI}$ satisfies the Williams–Beer axioms for measures of union information given in Definition 2.*


**Proof.** We address each of the axioms in turn.
1.Symmetry follows from the symmetry of mutual information, which in turn is a consequence of the well-known symmetry of joint entropy.2.Self-redundancy follows from the fact that agent *i* has access to $p(A_{i}|T)$ and p(T), which means that $p(A_{i},T)$ is one of the distributions in the set Q, which implies that $I_{\cup }\left(A_{i}\right)=I\left(A_{i};T\right)$.3.To show that monotonicity holds, begin by noting that
$$I\left(\overset{m}{\underset{i=1}{⋃}}A_{i};T\right)\geq I\left(\overset{m-1}{\underset{i=1}{⋃}}A_{i};T\right),$$
due to the monotonicity of mutual information. Let $Q_{m}$ be the set of distributions that the sources $A_{1},\ldots ,A_{m}$ can construct and $Q_{m-1}$ that which the sources $A_{1},\ldots ,A_{m-1}$ can construct. Since $Q_{m-1}\subseteq Q_{m}$, it is clear that
$$\max_{q\in Q_{m}}I_{q}\left(\overset{m}{\underset{i=1}{⋃}}A_{i};T\right)\geq \max_{q\in Q_{m-1}}I_{q}\left(\overset{m-1}{\underset{i=1}{⋃}}A_{i};T\right).$$Consequently,
$$\min\left\{I\left(\overset{m}{\underset{i=1}{⋃}}A_{i};T\right),\max_{q\in Q_{m}}I_{q}\left(\overset{m}{\underset{i=1}{⋃}}A_{i};T\right)\right\}\;\geq \min\left\{I\left(\overset{m-1}{\underset{i=1}{⋃}}A_{i};T\right),\;\max_{q\in Q_{m-1}}\;I_{q}\left(\overset{m-1}{\underset{i=1}{⋃}}A_{i};T\right)\right\},$$
which means monotonicity holds.4.Finally, the proof of equality for monotonicity is the same that was used above to show that the assumption that no source is a subset of another source entails no loss of generality. If $A_{m}\subseteq A_{m-1}$, then the presence of $A_{m}$ is irrelevant: $A=\overset{m}{\underset{i=1}{⋃}}A_{i}=\overset{m-1}{\underset{i=1}{⋃}}A_{i}$ and $Q_{m}=Q_{m-1}$, which implies that $I_{\cup }\left(A_{1},\ldots ,A_{m-1},A_{m}\right)=I_{\cup }\left(A_{1},\ldots ,A_{m-1}\right)$.
   □

### 4.2. Review of Suggested Properties: Griffith and Koch [16]

We now review properties of measures of union information and synergy that have been suggested in the literature in chronological order. The first set of properties was suggested by Griffith and Koch [16], with the first two being the following.

*Duplicating a predictor does not change synergistic information*; formally,
$$S\left(A_{1},\ldots ,A_{m}\to T\right)=S\left(A_{1},\ldots ,A_{m},A_{m+1}\to T\right),$$
where $A_{m+1}=A_{i}$, for some i=1,…,m. Griffith and Koch [16] show that this property holds if the equality for monotonicity property holds for the “corresponding” measure of union information (“corresponding” in the sense of Equation (Equation 7)). As shown in the previous subsection, $I_{\cup }^{\mathrm{CI}}$ satisfies this property, and so does the corresponding synergy $S^{\mathrm{CI}}$.*Adding a new predictor can decrease synergy*, which is a weak statement. We suggest a stronger property: *Following the monotonicity property, adding a new predictor cannot increase synergy*, which is formally written as
$$S\left(A_{1},\ldots ,A_{m}\to T\right)\geq S\left(A_{1},\ldots ,A_{m},A_{m+1}\to T\right).$$
This property simply follows from monotonicity for the corresponding measure of union information, which we proved above holds for $I_{\cup }^{\mathrm{CI}}$.

The next properties for any measure of union information were also suggested by Griffith and Koch [16]:1.Global positivity: $I_{\cup }\left(A_{1},\ldots ,A_{m}\right)\geq 0$.2.Self-redundancy: $I_{\cup }\left(A_{i}\right)=I\left(A_{i};T\right)$.3.Symmetry: $I_{\cup }\left(A_{1},\ldots ,A_{m}\right)$ is invariant under permutations of $A_{1},\ldots ,A_{m}$.4.Stronger monotonicity: $I_{\cup }\left(A_{1},\ldots ,A_{m}\right)\leq I_{\cup }\left(A_{1},\ldots ,A_{m},A_{m+1}\right)$, with equality if there is some $A_{i}$ such that $H(A_{m+1}|A_{i})=0$.5.Target monotonicity: for any (discrete) random variables *T* and *Z*, $I_{\cup }\left(A_{1},\ldots ,A_{m}\to T\right)\leq I_{\cup }\left(A_{1},\ldots ,A_{m}\to \left(T,Z\right)\right)$.6.Weak local positivity: for n=2 the derived partial informations are non-negative. This is equivalent to
$$\max\left\{I\left(Y_{1};T\right),I\left(Y_{2};T\right)\right\}\leq I_{\cup }\left(Y_{1},Y_{2}\right)\leq I\left(Y;T\right).$$7.Strong identity: $I_{\cup }\left(T\to T\right)=H\left(T\right)$.

We argued before that self-redundancy and symmetry are properties that follow trivially from a well-defined measure of union information [36]. In the following, we discuss in more detail properties 4 and 5, and return to the global positivity property later.

#### 4.2.1. Stronger Monotonicity

Property 4 in the above list was originally called monotonicity by Griffith and Koch [16]; we changed its name because we had already defined monotonicity in Definition 2, a weaker condition than stronger monotonicity. The proposed inequality clearly follows from the monotonicity of union information (the third Williams–Beer axiom). Now, if there is some $A_{i}$ such that $H(A_{m+1}|A_{i})=0$ (equivalently, if $A_{m+1}$ is a deterministic function of $A_{i}$), Griffith and Koch [16] suggest that we must have equality. Recall Axiom 4 (equality for monotonicity) in the extension of the WB axioms (Definition 2). It states that equality must hold if $A_{m}\subseteq A_{m-1}$. In this context, $A_{m}$ and $A_{m-1}$ are sets of random variables, for example, $A_{m}=\left\{Y_{1},Y_{2}\right\}$ and $A_{m-1}=\left\{Y_{1},Y_{2},Y_{3}\right\}$. There is a different point of view we may take. The only way that $A_{m}$ is a subset of $A_{m-1}$ is if $A_{m}$, when viewed as a random vector (in this case, write $A_{m}=\left(Y_{1},Y_{2}\right)$ and $A_{m-1}=\left(Y_{1},Y_{2},Y_{3}\right)$), is a subvector of $A_{m-1}$. A subvector of a random vector is a deterministic function, and no information gain can come from applying a deterministic function to a random vector. As such, there is no information gain when one considers $A_{m}$, a function of $A_{m-1}$, if one already has access to $A_{m-1}$. Griffith and Koch [16] argue similarly, there is no information gain by considering $A_{m+1}$—a function of $A_{i}$—in addition to $A_{i}$. In conclusion, considering the ‘equality for monotonicity’ strictly through a set-inclusion perspective, stronger monotonicity does not follow. On the other hand, extending the idea of set inclusion to the more general context of functions of random variables, then stronger monotonicity follows, because $\left\{A_{m+1}\right\}=\left\{f\left(A_{i}\right)\right\}$ is a subset of $\left\{A_{i}\right\}$, hence there is no information gain by considering $A_{m+1}=f\left(A_{i}\right)$ in addition to $A_{i}$. As such, we obtain $I_{\cup }\left(A_{1},\ldots ,A_{m}\right)=I_{\cup }\left(A_{1},\ldots ,A_{m},A_{m+1}\right)$. Consequently, it is clear that stronger monotonicity must hold for any measure of union information. We, thus, argue in favor of extending the concept of subset inclusion of property 4 in Definition 2 to include the concept of deterministic functions.

#### 4.2.2. Target Monotonicity

Let us move on to target monotonicity, which we argue should not hold in general. This precise same property was suggested, but for a measure of redundant information, by Bertschinger et al. [19]; they argue that a measure of redundant information should satisfy
$$I_{\cap }\left(A_{1},\ldots ,A_{m}\to T\right)\leq I_{\cap }\left(A_{1},\ldots ,A_{m}\to \left(T,Z\right)\right),$$
for any discrete random variable *Z*, as they argue that this property *‘captures the intuition that if $A_{1},\ldots ,A_{m}$ share some information about T, then at least the same amount of information is available to reduce the uncertainty about the joint outcome of (T,Z)’*. Since most PID approaches have been built upon measures of redundant information, it is simpler to refute this property. Consider $I_{\cap }^{d}$, which we argue is one of the most well-motivated and accepted measures of redundant information (as defined in (Equation 2)): it satisfies the WB axioms, it is based on the famous Blackwell channel preorder—thus inheriting a well-defined and rigorous operation interpretation—it is based on channels, just as I(X;T) was originally motivated based on channels, and is defined for any number of input variables, which is more than most PID measures can accomplish. Consider also the distribution presented in Table 4, which satisfies $T=Y_{1}\;\mathrm{AND}\;Y_{2}$ and $Z=(Y_{1},Y_{2})$.

From a game theory perspective, since neither agent ($Y_{1}$ or $Y_{2}$) has an advantage when predicting *T* (because the channels that each agent has access to have the same conditional distributions), neither agent has any unique information. Moreover, redundancy—as computed by $I_{\cap }^{d}\left(Y_{1},Y_{2}\to T\right)$—evaluates to approximately 0.311. However, when considering the pair (T,Z), the structure that was present in *T* is now destroyed, in the sense that now there is no degradation order between the channels that each agent has access to. Note that $p(\left(t,z\right),y_{1},y_{2})$ is a relabeling of the COPY distribution. As such, $I_{\cap }^{d}\left(Y_{1},Y_{2}\to \left(T,Z\right)\right)=0<I_{\cap }^{d}\left(Y_{1},Y_{2}\to T\right)$, contradicting the property proposed by Bertschinger et al. [19].

For a similar reason, we believe that this property should not hold for a general measure of union information, even if the measure satisfies the extension of the Williams–Beer axioms, as our proposed measure does. For instance, consider the distribution presented in Table 5.

This distribution yields $I_{\cup }^{\mathrm{CI}}\left(Y_{1},Y_{2}\to T\right)\approx 0.91>0.90\approx I_{\cup }^{\mathrm{CI}}\left(Y_{1},Y_{2}\to \left(T,Z\right)\right)$, meaning target monotonicity does not hold. This happens because although $I_{p}\left(Y;T\right)\leq I_{p}\left(Y;T,Z\right)$, it is not necessarily true that $I_{q}\left(Y;T\right)\leq I_{q}\left(Y;T,Z\right)$. The union information measure derived from the degradation order between channels, defined as the ‘dual’ of (Equation 2), also agrees with our conclusion [22]. For the distribution in Table 4 we have $I_{\cup }^{d}\left(Y_{1},Y_{2}\to T\right)\approx 0.331>0=I_{\cup }^{d}\left(Y_{1},Y_{2}\to \left(T,Z\right)\right)$, for the same reason as above: considering (T,Z) as the target variable destroys the structure present in *T*. We agree with the remaining properties suggested by Griffith and Koch [16] and we will address those later.

### 4.3. Review of Suggested Properties: Quax et al. [37]

Moving on to additional properties, Quax et al. [37] suggest the following properties for a measure of synergy:1.Non-negativity: $S(A_{1},\ldots ,A_{m}\to T)\geq 0$.2.Upper-bounded by mutual information: S(Y→T)≤I(Y;T).3.Weak symmetry: $S(A_{1},\ldots ,A_{m}\to T)$ is invariant under any reordering of $A_{1},\ldots ,A_{m}$.4.Zero synergy about a single variable: $S(Y_{i}\to T)=0$ for any i∈{1,…,n}.5.Zero synergy in a single variable: $S(Y\to Y_{i})=0$ for any i∈{1,…,n}.

Let us comment on the proposed ‘zero synergy’ properties (4 and 5) under the context of PID. Property 4 seems to have been proposed with the rationale that synergy can only exist for at least two sources, which intuitively makes sense, as synergy is often defined as ‘the information that is present in the pair, but that is not retrievable from any individual variable’. However, because of the way a synergy-based PID is constructed—or weak-synergy, as Gutknecht et al. [36] call it—synergy must be defined as in (Equation 7), so that, for example, in the bivariate case, $S\left(Y_{1}\to T\right):=I\left(Y;T\right)-I_{\cup }\left(Y_{1}\to T\right)=I\left(Y_{2};T|Y_{1}\right)$, because of self-redundancy of union information and the chain rule of mutual information [12], and since $I(Y_{2};T|Y_{1})$ is in general larger than 0, we reject the property ‘zero synergy about a single variable’. We note that our rejection of this property is based on the PID perspective. There may be other areas of research where it makes sense to demand that any (single) random variable alone has no synergy about any target, but under the PID framework, this must not happen, particularly so that we obtain a valid information decomposition.

Property 5, ‘zero synergy in a single variable’, on the other hand, must hold because of self-redundancy. That is because, for any i∈{1,…,n}, $S\left(Y\to Y_{i}\right):=I\left(Y;Y_{i}\right)-I_{\cup }\left(Y\to Y_{i}\right)=I\left(Y_{i};Y_{i}\right)-I\left(Y;Y_{i}\right)=H\left(Y_{i}\right)-H\left(Y_{i}\right)=0$.

### 4.4. Review of Suggested Properties: Rosas et al. [38]

Based on the proposals of Griffith et al. [24], Rosas et al. [38] suggested the following properties for a measure of synergy:Target data processing inequality: if $Y-T_{1}-T_{2}$ is a Markov chain, then $S\left(Y\to T_{1}\right)\geq S\left(Y\to T_{2}\right)$.Channel convexity: S(Y→T) is a convex function of P(T|Y) for a given P(Y).

We argue that a principled measure of synergy does not need to satisfy these properties (in general). Consider the distribution presented in Table 6, in which $T_{1}$ is a relabeling of the COPY distribution and $T_{2}=Y_{1}\;\mathrm{xor}\;Y_{2}$.

Start by noting that since $T_{2}$ is a deterministic function of $T_{1}$, then $Y-T_{1}-T_{2}$ is a Markov chain. Since $Y_{1}⊥Y_{2}|T_{1}$, our measure $S^{\mathrm{CI}}\left(Y_{1},Y_{2}\to T_{1}\right)=I\left(Y;T_{1}\right)-I_{\cup }^{\mathrm{CI}}\left(Y_{1},Y_{2}\to T_{1}\right)=0$ leads to zero synergy. On the other hand, $S^{\mathrm{CI}}\left(Y_{1},Y_{2}\to T_{2}\right)=1$, contradicting the first property suggested by Rosas et al. [38]. This happens because Y1⊥pY2|T2, so synergy is positive. The loss of conditional independence of the inputs (given the target) when one goes from considering the target $T_{1}$ to $T_{2}$ is the reason why synergy increases. It can be easily seen that $S^{d}$, the measure of synergy derived from Kolchinsky’s proposed union information measure $I_{\cup }^{d}$ [22], agrees with this. A simpler way to see this is by noticing that the XOR distribution must yield 1 bit of synergistic information, and many PID measures do not yield 1 bit of synergistic information for the COPY distribution.

The second suggested property argues that synergy should be a convex function of P(T|Y), for fixed P(Y). Our measure of synergy does not satisfy this property, even though it is derived from a measure of union information that satisfies the extension of the WB axioms. For instance, consider the XOR distribution with one extra outcome. We introduce it in Table 7 and parameterize it using r=p(T=0|Y=(0,0))∈[0,1]. Notice that this modification does not affect P(Y).

Synergy, as measured by $S^{\mathrm{CI}}\left(Y_{1},Y_{2}\to T\right)$, is maximized when *r* equals 1 (the distribution becomes the standard XOR) and minimized when *r* equals 0. We do not see an immediate reason as to why a general synergy function should be convex in p(t|y), or why it should have a unique minimizer as a function of *r*. Recall that a function *S* is convex if $\forall t\in \left[0,1\right],\forall x_{1},x_{2}\in D$, we have
$$S\left(tx_{1}+\left(1-t\right)x_{2}\right)\leq tS\left(x_{1}\right)+\left(1-t\right)S\left(x_{2}\right).$$

In the following, we slightly abuse the notation of the input variables of a synergy function. Our synergy measure $S^{\mathrm{CI}}$, when considered as a function of *r*, does not satisfy this inequality. For the adapted XOR distribution, take t=0.5, $x_{1}=0$, and $x_{2}=0.5$. We have
$$S^{\mathrm{CI}}\left(0.5\times 0+0.5\times 0.5\right)=S^{\mathrm{CI}}\left(0.25\right)\approx 0.552$$
and
$$0.5\times S^{\mathrm{CI}}\left(0\right)+0.5\times S^{\mathrm{CI}}\left(0.5\right)\approx 0.5\times 0.270+0.5\times 0.610\approx 0.440,$$
contradicting the property of channel convexity. $S^{d}$ agrees with this. We slightly change p(y) in the above distribution to obtain a new distribution, which we present in Table 8.

This distribution does not satisfy the convexity inequality, since
$$S^{d}\left(0.5\times 0+0.5\times 0.5\right)\approx 0.338>0.3095\approx 0.5S^{d}\left(0\right)+0.5S^{d}\left(0.5\right).$$
This can be easily seen since $K^{\left(1\right)}=K^{\left(2\right)}$ for any r∈[0,1], hence we may choose $K^{Q}=K^{\left(1\right)}$ to compute $S^{d}=I\left(Y;T\right)-I\left(Q;T\right)$, which is not convex for this particular distribution. To conclude this section, we present a plot of $S^{\mathrm{CI}}\left(Y_{1},Y_{2}\right)$ and $S^{d}\left(Y_{1},Y_{2}\right)$ as a function of *r* in Figure 3, for the distribution presented in Table 7.

With this we do not mean that these properties should not hold for an arbitrary PID. We are simply showing that some properties must be satisfied in the context of PID (such as the WB axioms), whereas other properties are not necessary for PID (such as the previous two properties).

### 4.5. Relationship with the Extended Williams–Beer Axioms

We now prove which of the introduced properties are implied by the extension of the Williams–Beer axioms for measures of union information. In what follows, assume that the goal is to decompose the information present in the distribution $p\left(y,t\right)=p\left(y_{1},\ldots ,y_{n},t\right)$.

**Theorem 2.** 
*Let $I_{\cup }$ be a measure of union information that satisfies the extension of the Williams–Beer axioms (symmetry, self-redundancy, monotonicity, and equality for monotonicity) for measures of union information as in Definition (2). Then, $I_{\cup }$ also satisfies the following properties of Griffith and Koch [16]: global positivity, weak local positivity, strong identity, “duplicating a predictor does not change synergistic information”, and “adding a new predictor cannot increase synergy”.*


**Proof.** We argued before that the last two properties follow from the definition of a measure of union information. Global positivity is a direct consequence of monotonicity and the non-negativity of mutual information: $I_{\cup }\left(A_{1},\ldots ,A_{m}\right)\geq I_{\cup }\left(A_{1}\right)=I\left(A_{1};T\right)\geq 0$.Weak local positivity holds because monotonicity and self-redundancy imply that $I_{\cup }\left(Y_{1},Y_{2}\right)\geq I_{\cup }\left(Y_{1}\right)=I\left(Y_{1};T\right)$, as well as $I_{\cup }\left(Y_{1},Y_{2}\right)\geq I\left(Y_{2};T\right)$, hence $\max\{I\left(Y_{1};T\right),I\left(Y_{2};T\right)\}$$\leq I_{\cup }\left(Y_{1},Y_{2}\right)$. Moreover, $I_{\cup }\left(Y_{1},Y_{2}\right)\leq I_{\cup }\left(Y_{1},Y_{2},Y_{12}\right)=I_{\cup }\left(Y_{12}\right)=I\left(Y;T\right)$.Strong identity follows trivially from self-redundancy, since $I_{\cup }\left(T\to T\right)=I\left(T;T\right)=H\left(T\right)$.    □

**Theorem 3.** 
*Consider a measure of union information that satisfies the conditions of Theorem 2. If synergy is defined as in Equation (Equation 7), it satisfies the following properties of [37]: non-negativity, upper-bounded by mutual information, weak symmetry, and zero synergy in a single variable.*


**Proof.** Non-negativity of synergy and upper-bounded by mutual information follow from the definition of synergy and from the fact that for whichever source $\left(A_{1},\ldots ,A_{m}\right)$, with m≥1, we have that $I_{\cup }\left(A_{1},\ldots ,A_{m}\to T\right)\leq I\left(Y;T\right)$.Weak symmetry follows trivially from the fact that both $I\left(Y;T\right)$ and $I_{\cup }\left(A_{1},\ldots ,A_{m}\to T\right)$ are symmetric in the relevant arguments.Finally, zero synergy in a single variable follows from self-redundancy together with the definition of synergy, as shown above.    □

## 5. Previous Measures of Union Information and Synergy

We now review other measures of union information and synergy proposed in the literature. For the sake of brevity, we will not recall all their definitions, only some important conclusions. We suggest the interested reader consult the bibliography for more information.

### 5.1. Qualitative Comparison

Griffith and Koch [16] review three previous measures of synergy:$S^{\mathrm{WB}}$, derived from $I_{\cap }^{\mathrm{WB}}$, the original redundancy measure proposed by Williams and Beer [1], using the IEP;the *whole-minus-sum* (WMS) synergy, $S^{\mathrm{WMS}}$;the *correlational importance* synergy, $S^{\Delta I}$.

These synergies can be interpreted as resulting directly from measures of union information; that is, they are explicitly written as $S\left(\alpha \to T\right)=I\left(Y;T\right)-I_{\cup }\left(\alpha \to T\right)$, where $I_{\cup }$ may not necessarily satisfy our intuitions of a measure of union information, as in Definition 2, except for $S^{\Delta I}$, which has the form of a Kullback–Leibler divergence.

Griffith and Koch [16] argue that $S^{\mathrm{WB}}$*overestimates* synergy, which is not a surprise, as many authors criticized $I_{\cap }^{\mathrm{WB}}$ for not measuring informational content, only informational values [13]. The WMS synergy, on the other hand, which can be written as a difference of total correlations, can be shown to be equal to the difference between synergy and redundancy for n=2, which is not what is desired in a measure of synergy. For n>2, the authors show that the problem becomes even more exacerbated: $S^{\mathrm{WMS}}$ equals synergy minus the redundancy *counted multiple times*, which is why the authors argue that $S^{\mathrm{WMS}}$*underestimates* synergy. Correlational importance, $S^{\Delta I}$, is known to be larger than I(Y;T) for some distributions, excluding it from being an appropriately interpretable measure of synergy.

Faced with these limitations, Griffith and Koch [16] introduce their measure of union information, which they define as
(9)$$\begin{array}{cc} I_{\cup }^{\mathrm{VK}}\left(A_{1},\ldots ,A_{m}\to T\right):= & \min_{p^{*}}I_{p^{*}}\left(\overset{m}{\underset{i=1}{⋃}}A_{i};T\right)\\ \; & s.t.p^{*}\left(A_{i},T\right)=p\left(A_{i},T\right),i=1,\ldots ,m, \end{array}$$
where the minimization is over joint distributions of $A_{1},\ldots ,A_{m},T$, alongside the derived measure of synergy $S^{\mathrm{VK}}\left(\alpha \to T\right)=I\left(Y;T\right)-I_{\cup }^{\mathrm{VK}}\left(\alpha \to T\right)$. This measure quantifies union information as the least amount of information that source α has about *T* when the source–target marginals (as determined by α) are fixed by *p*. Griffith and Koch [16] also established the following inequalities for the synergistic measures they reviewed:(10)$$\begin{array}{c} \max\left\{0,S^{\mathrm{WMS}}\left(\alpha \to T\right)\right\}\leq S^{\mathrm{VK}}\left(\alpha \to T\right)\leq S^{\mathrm{WB}}\left(\alpha \to T\right)\leq I\left(\overset{m}{\underset{i=1}{⋃}}A_{i};T\right), \end{array}$$
where $\alpha =(A_{1},\ldots ,A_{m})$. At the time, Griffith and Koch [16] did not provide a way to analytically compute their measure. Later, Kolchinsky [22] showed that the measure of union information derived from the degradation order, $I_{\cup }^{d}$, is equivalent to $I_{\cup }^{\mathrm{VK}}$, and provided a way to compute it. For this reason, we will only consider $I_{\cup }^{d}$.

After the work of Griffith and Koch [16] in 2014, we are aware of only three other suggested measures of synergy:$S^{\mathrm{MSRV}}$, proposed by Quax et al. [37], where MSRV stands for *maximally synergistic random variable*;*synergistic disclosure*, $S^{\mathrm{SD}}$, proposed by Rosas et al. [38];$S^{d}$, proposed by Kolchinsky [22].

The first two proposals do not define synergy via a measure of union information. They define synergy through an auxiliary random variable, *Z*, which has positive information about the whole—that is, I(Z;Y)>0—but no information about any of the parts—that is, $I(Z;Y_{i})=0,i=1,\ldots ,n$. While this property has an appealing operational interpretation, we believe that it is too restrictive; that is, we believe that information *can* be synergistic, even if it provides some positive information about some part of *Y*.

The authors of $S^{\mathrm{MSRV}}$ show that their proposed measure is incompatible with PID and that it cannot be computed for all distributions, as it requires the ability to compute orthogonal random variables, which is not always possible [37]. A counter-intuitive example for the value of this measure can be seen for the AND distribution, defined by $T=Y_{1}\;\mathrm{AND}\;Y_{2}$, with $Y_{1}$ and $Y_{2}$ i.i.d. taking values in {0,1} with equal probability. In this case, $S^{\mathrm{MSRV}}=0.5$, a value that we argue is too large, because whenever $Y_{1}$ (respectively $Y_{2}$) is 0, then *T* does not depend on $Y_{2}$ (respectively $Y_{1}$) (which happens with probability 0.75). Consequently, $S^{\mathrm{MSRV}}/I\left(Y;T\right)\approx 0.5/0.811\approx 0.617$ may be too large of a synergy ratio for this distribution. As the authors note, the only other measure that agrees with $S^{\mathrm{MSRV}}$ for the AND distribution is $S^{\mathrm{WB}}$, which Griffith and Koch [16] argued also overestimates synergy.

Concerning $S^{\mathrm{SD}}$, we do not have any criticism, except for the one already pointed out by Gutknecht et al. [36]: they note that the resulting decomposition from $S^{\mathrm{SD}}$ is not a standard PID, in the sense that it does not satisfy a consistency equation (see [36] for more details), which implies that ‘… the atoms cannot be interpreted in terms of parthood relations with respect to mutual information terms …. For example, we do not obtain any atoms interpretable as unique or redundant information in the case of two sources’ [36]. Gutknecht et al. [36] suggest a very simple modification to the measure so that it satisfies the consistency equation.

For the AND distribution, $S^{\mathrm{SD}}$ evaluates to approximately 0.311, as does $S^{d}$, whereas our measure yields $S^{\mathrm{CI}}\approx 0.270$, as the information that the parts cannot obtain when they combine their marginals, under distribution *q*. This shows that these four measures are not equivalent.

### 5.2. Quantitative Comparison

Griffith and Koch [16] applied the synergy measures they reviewed to other distributions. We show their results below in Table 9 and compare them with the synergy resulting from our measure of union information, $S^{\mathrm{CI}}$, with the measure of Rosas et al. [38], $S^{\mathrm{SD}}$, and that of Kolchinsky [22], $S^{d}$. Since the code for the computation of $S^{\mathrm{MSRV}}$ is no longer available online, we do not present it.

We already saw the definition of the AND, COPY, and XOR distributions. The XORDUPLICATE and ANDDUPLICATE are built from the XOR and the AND distributions by inserting a duplicate source variable $Y_{3}=Y_{2}$. The goal is to test if the presence of a duplicate predictor impacts the different synergy measures. The definitions of the remaining distributions are presented in Appendix A. Some of these are trivariate, and for those we compute synergy as
(11)$$\begin{array}{c} S\left(Y_{1},Y_{2},Y_{3}\to T\right)=I\left(Y_{1},Y_{2},Y_{3};T\right)-I_{\cup }\left(Y_{1},Y_{2},Y_{3}\to T\right), \end{array}$$
unless the synergy measure is directly defined (as opposed to being defined via a union information measure). We now comment on the results. It should be noted that Kolchinsky [22] suggested that unique information $U_{1}$ and $U_{2}$ should be computed from measures of redundant information, and excluded information $E_{1}$ and $E_{2}$ should be computed from measures of union information, as in our case. However, since we will only present the decompositions for the bivariate case and in this case $E_{1}=U_{2}$ and $E_{2}=U_{1}$, we present the results considering unique information, as is mostly performed in the literature.

XOR yields I(Y;T)=1. The XOR distribution is the hallmark of synergy. Indeed, the only solution of (Equation 1) is $\left(S,R,U_{1},U_{2}\right)=\left(1,0,0,0\right)$, and all of the above measures yield 1 bit of synergy.AND yields I(Y;T)≈0.811. Unlike XOR, there are multiple solutions for (Equation 1), and none is universally agreed upon, since different information measures capture different concepts of information.COPY yields I(Y;T)=2. Most PID measures argue one of two different possibilities for this distribution. They suggest that the solution is either $\left(S,R,U_{1},U_{2}\right)=\left(1,1,0,0\right)$ or (0,0,1,1). Our measure suggests that all information flows uniquely from each source.RDNXOR yields I(Y;T)=2. In words, this distribution is the concatenation of two XOR ‘blocks’, each of which has its own symbols, and not allowing the two blocks to mix. That is, both $Y_{1}$ and $Y_{2}$ can determine in which XOR block the resulting value *T* will be—which intuitively means that they both have this information, meaning it is redundant—but neither $Y_{1}$ nor $Y_{2}$ have information about the outcome of the XOR operation—as is expected in the XOR distribution—which intuitively means that such information must be synergistic. All measures except $S^{\mathrm{WMS}}$ agree with this.RDNUNQXOR yields I(Y;T)=4. According to Griffith and Koch [16], it was constructed to carry 1 bit of each information type. Although the solution is not unique, it must satisfy $U_{1}=U_{2}$. Indeed, our measure yields the solution $\left(S,R,U_{1},U_{2}\right)=\left(1,1,1,1\right)$, like most measures except $S^{\mathrm{WB}}$ and $S^{\mathrm{WMS}}$. This confirms the intuition by Griffith and Koch [16] that $S^{\mathrm{WB}}$ and $S^{\mathrm{WMS}}$ overestimate and underestimate synergy, respectively. In fact, in the decomposition resulting from $S^{\mathrm{WB}}$, there are 2 bits of synergy and 2 bits of redundancy, which we argue cannot be the case, as this would imply that $U_{1}=U_{2}=0$, and given the construction of this distribution, it is clear that there is some unique information since, unlike in RDNXOR, the XOR blocks are allowed to mix, thus $\left(T,Y_{1},Y_{2}\right)=\left(1,0,1\right)$ is a possible outcome, but so is $\left(T,Y_{1},Y_{2}\right)=\left(2,0,2\right)$. That is not the case with RDNXOR. On the other hand, $S^{\mathrm{WMS}}$ yields zero synergy and redundancy, with $U_{1}$ and $U_{2}$ each evaluating to 2 bits. Since this distribution is a mix of blocks satisfying a relation of the form $T=Y_{1}\;\mathrm{xor}\;Y_{2}$, we argue that there must be some non-null amount of synergy, which is why we claim that $S^{\mathrm{WMS}}$ is not valid.XORDUPLICATE yields I(Y;T)=1. All measures correctly identify that the duplication of a source should not change synergy, at least for this particular distribution.ANDDUPLICATE yields I(Y;T)≈0.811. Unlike in the previous example, both $S^{\mathrm{WMS}}$ and $S^{\Delta I}$ yield a change in their synergy value. This is a shortcoming, since duplicating a source should not increase either synergy or union information. The other measures are not affected by the duplication of a source.XORLOSES yields I(Y;T)=1. Its distribution is the same as XOR but with a new source $Y_{3}$ satisfying $T=Y_{3}$. As such, since $Y_{3}$ uniquely determines *T*, we expect no synergy. All measures agree with this.XORMULTICOAL yields I(Y;T)=1. Its distribution is such that any pair $\left(Y_{i},Y_{j}\right)$, i,j=1,2,3,i≠j is able to determine *T* with no uncertainty. All measures agree that the information present in this distribution is purely synergistic.

From these results, we agree with Griffith and Koch [16] that $S^{\mathrm{WB}}$, $S^{\mathrm{WMS}}$, and $S^{\Delta I}$ are not good measures of synergy: they do not satisfy many of our intuitions and overestimate synergy, not being invariant to duplicate sources or taking negative values. For these reasons, and those presented in Section 5, we reject those measures of synergy. In the next section, we comment on the remaining measures $S^{d}$, $S^{\mathrm{SD}}$, and $S^{\mathrm{CI}}$.

### 5.3. Relation to Other PID Measures

Kolchinsky [22] introduced $I_{\cup }^{d}$ and showed that this measure is equivalent to $I_{\cup }^{\mathrm{VK}}$ [16] and to $I_{\cup }^{\mathrm{BROJA}}$ [15], in the sense that the three of them achieve the same optimum value [22]. The multivariate extension of $I_{\cup }^{\mathrm{BROJA}}$ was proposed by Griffith and Koch [16], defined as
$$I_{\cup }^{\mathrm{BROJA}}\left(A_{1},\ldots ,A_{m}\to T\right):=\min_{\tilde{A_{1}},\ldots ,\tilde{A_{m}}}I\left(\tilde{A_{1}},\ldots ,\tilde{A_{m}};T\right)\;\;\mathrm{such}\;\mathrm{that}\;\;\forall i\;\;P\left(\tilde{A_{i}},T\right)=P\left(A_{i},T\right),$$
which we present because it makes it clear what conditions are enforced upon the marginals. There is a relation between $I_{\cup }^{\mathrm{BROJA}}\left(A_{1},\ldots ,A_{m}\right)=I_{\cup }^{d}\left(A_{1},\ldots ,A_{m}\right)$ and $I_{\cup }^{\mathrm{CI}}\left(A_{1},\ldots ,A_{m}\right)$ whenever the sources $A_{1},\ldots A_{m}$ are singletons. In this case, and only in this case, the set Q involved in the computation of $I_{\cup }^{\mathrm{CI}}\left(A_{1},\ldots ,A_{m}\right)$ has only one element: $q\left(t,a_{1},\ldots ,a_{m}\right)=p\left(t\right)p\left(a_{1}|t\right)\ldots p\left(a_{m}|t\right)$. Since this distribution, as well as the original distribution *p*, are both admissible points in $I_{\cup }^{d}$, we have that $I_{\cup }^{d}\leq I_{\cup }^{\mathrm{CI}}$, which implies that $S^{d}\geq S^{\mathrm{CI}}$. On the other hand, if there is at least one source $A_{1},\ldots ,A_{m}$ that is not a singleton, the measures are not trivially comparable. For example, suppose we wish to compute $I_{\cup }\left(\left(Y_{1},Y_{2}\right),\left(Y_{2},Y_{3}\right)\right)$. We know that the solution of $I_{\cup }^{d}\left(\left(Y_{1},Y_{2}\right),\left(Y_{2},Y_{3}\right)\right)$ is a distribution $p^{*}$ whose marginals $p^{*}\left(y_{1},y_{2},t\right)$ and $p^{*}\left(y_{2},y_{3},t\right)$ must coincide with the marginals under the original *p*. However, in the computation of $I_{\cup }^{\mathrm{CI}}\left(\left(Y_{1},Y_{2}\right),\left(Y_{2},Y_{3}\right)\right)$, it may be the case that the solution $p^{*}$ of $I_{\cup }^{d}\left(\left(Y_{1},Y_{2}\right),\left(Y_{2},Y_{3}\right)\right)$ is not in the set Q, involved in the computation of $I_{\cup }^{\mathrm{CI}}$, and it achieves a lower mutual information with *T*. That is, it might be the case that $I_{p^{*}}\left(Y;T\right)<I_{q}\left(Y;T\right)$, for all q∈Q. In such a case, we would have $I_{\cup }^{d}>I_{\cup }^{\mathrm{CI}}$.

It is convenient to be able to upper-bound certain measures with other measures. For example, Gomes and Figueiredo [26] (see that paper for the definitions of these measures) showed that for any source $(A_{1},\ldots ,A_{m}),m\geq 1$,
$$I_{\cap }^{d}\left(A_{1},\ldots ,A_{m}\right)\leq I_{\cap }^{\ln}\left(A_{1},\ldots ,A_{m}\right)\leq I_{\cap }^{\mathrm{mc}}\left(A_{1},\ldots ,A_{m}\right).$$
However, we argue that the inability to draw such strong conclusions (or bounds) is a positive aspect of PID. This is because there are many different ways to define the information (be it redundant, unique, union, etc.) that one wishes to capture. If one could trivially relate all measures, it would mean that it would be possible to know *a priori* how those measures would behave. Consequently, this would imply the absence of variability/freedom in how to measure different information concepts, as those measures would capture, non-equivalent but similar types of information, as they would all be ordered. It is precisely because one cannot order different measures of information trivially that PID provides a rich and complex framework to distinguish different types of information, although we believe that PID is still in its infancy.

James et al. [17] introduced a measure of unique information, which we recall now. In the bivariate case—i.e., consider $p(y_{1},y_{2},t)$—let *q* be the maximum entropy distribution that preserves the marginals $p(y_{1},t)$ and $p(y_{2},t)$, and let *r* be the maximum entropy distribution that preserves the marginals $p(y_{1},t)$, $p(y_{2},t)$, and $p(y_{1},y_{2})$. Although there is no closed form for *r*, which has to be computed using an iterative algorithm [39], it may be shown that the solution for *q* is $q\left(y_{1},y_{2},t\right)=p\left(t\right)p\left(y_{1}|t\right)p\left(y_{2}|t\right)$ (see, e.g., [17]). This is the same distribution *q* that we consider for the bivariate decomposition (Equation 3). James et al. [17] suggest defining unique information $U_{i}$ as the least change (in sources–target mutual information) that involves the addition of the $\left(Y_{i},T\right)$ marginal constraint, that is,
(12)$$\begin{array}{c} U_{1}=\min\left\{I_{q}\left(Y_{1};T|Y_{2}\right),I_{r}\left(Y_{1};T|Y_{2}\right)\right\}, \end{array}$$
and analogously for $U_{2}$. They show that their measure yields a non-negative decomposition for the bivariate case. Since $I\left(Y_{1};T|Y_{2}\right)=S+U_{1}$, some algebra leads to
(13)$$\begin{array}{c} S^{\mathrm{dep}}=I\left(Y;T\right)-\min\left\{I_{q}\left(Y;T\right),I_{r}\left(Y;T\right)\right\}, \end{array}$$
where $S^{\mathrm{dep}}$ is the synergy resulting from the decomposition of James et al. [17] in the bivariate case. Recall that our measure of synergy for the bivariate case is given by
(14)$$\begin{array}{c} S^{\mathrm{CI}}=I\left(Y;T\right)-\min\left\{I_{q}\left(Y;T\right),I_{p}\left(Y;T\right)\right\}. \end{array}$$
The similarity is striking. Computing $S^{\mathrm{dep}}$ for the bivariate distributions in Table 9 shows that it coincides with the decomposition given by our measure, although this is not the case in general. We could not obtain $S^{\mathrm{dep}}$ for the RDNUNQXOR distribution because the algorithm that computes *r* did not finish in the allotted time of 10 min. James et al. [17] showed that for whichever bivariate distribution $I_{r}\left(Y;T\right)\leq I_{p}\left(Y;T\right)$; therefore, for the bivariate case we have $S^{\mathrm{CI}}\leq S^{\mathrm{dep}}$. Unfortunately, the measure of unique information proposed by James et al. [17], unlike the usual proposals of intersection or union information, does not allow for the computation of the partial information atoms in the complete redundancy lattice if n>2. The authors also comment that it is not clear if their measure satisfies monotonicity when n>2. Naturally, our measure is not the same as $S^{\mathrm{dep}}$, so it does not retain the operational interpretation of unique information $U_{i}$ being the least amount that influences I(Y;T) when the marginal constraint $\left(Y_{i},T\right)$ is added to the resulting maximum entropy distributions. Given the form of $S^{\mathrm{dep}}$, one could define $I_{\cup }^{\mathrm{dep}}:=\min\left\{I_{q}\left(Y;T\right),I_{r}\left(Y;T\right)\right\}$ and study its properties. Clearly, it does not satisfy the self-redundancy axiom, but we wonder if it could be adjusted so that it satisfies all of the proposed axioms. The n=2 decomposition retains the operational interpretation of the original measure, but it is not clear whether this is true for n>2. For the latter case, the maximum entropy distributions that we wrote as *q* and *r* have different definitions [17]. We leave this for future work.

## 6. Conclusions and Future Work

In this paper, we introduced a new measure of *union information* for the *partial information decomposition* (PID) framework, based on the channel perspective, which quantifies synergy as the information that is beyond conditional independence of the sources, given the target. This measure has a clear interpretation and is very easy to compute, unlike most measures of union information or synergy, which require solving an optimization problem. The main contributions and conclusions of the paper can be summarized as follows.

We introduced new measures of union information and synergy for the PID framework, which thus far was mainly developed based on measures of redundant or unique information. We provided its operational interpretation and defined it for an arbitrary number of sources.We proposed an extension of the Williams–Beer axioms for measures of union information and showed our proposed measure satisfies them.We reviewed, commented on, and rejected some of the previously proposed properties for measures of union information and synergy in the literature.We showed that measures of union information that satisfy the extension of the Williams–Beer axioms necessarily satisfy a few other appealing properties, as well as the derived measures of synergy.We reviewed previous measures of union information and synergy, critiqued them, and compared them with our proposed measure.The proposed conditional independence measure is very simple to compute.We provide code for the computation of our measure for the bivariate case and for source $\left\{\left\{Y_{1}\right\},\left\{Y_{2}\right\},\left\{Y_{3}\right\}\right\}$ in the trivariate case.

Finally, we believe this paper opens several avenues for future research, thus we point out several directions to be pursued in upcoming work:We saw that the synergy yielded by the measure of James et al. [17] is given by $S^{\mathrm{dep}}=I\left(Y;T\right)-\min\left\{I_{q}\left(Y;T\right),I_{r}\left(Y;T\right)\right\}$. Given its analytical expression, one could start by defining a measure of union information as $I_{\cup }\left(Y_{1},Y_{2}\to T\right)=\min\left\{I_{q}\left(Y;T\right),I_{r}\left(Y;T\right)\right\}$, possibly tweak it so it satisfies the WB axioms, study its properties, and possibly extend it to the multivariate case.Our proposed measure may ignore conditional dependencies that are present in *p* in favor of maximizing mutual information, as we commented in Section 3.3. This is a compromise so that the measure satisfies monotonicity. We believe this is a potential drawback of our measure, and we suggest the investigation of a measure similar to ours, but that does not ignore conditional dependencies that it has access to.Extending this measure for absolutely continuous random variables.Implementing our measure in the dit package [40].This paper reviewed measures of union information and synergy, as well as properties that were suggested throughout the literature. Sometimes this was by providing examples where the suggested properties fail, and other times simply by commenting. We suggest performing something similar for measures of redundant information.

## 7. Code Availability

The code is publicly available at https://github.com/andrefcorreiagomes/CIsynergy/ and requires the dit package [40] (accessed on 12 January 2024).

## Figures and Tables

**Figure 1 entropy-26-00271-f001:**
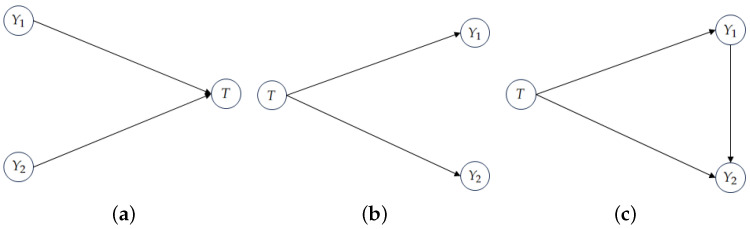
(**a**) Assuming faithfulness [28], this is the only three-variable directed acyclic graph (DAG) that satisfies $Y_{1}⊥Y_{2}$ and Y1⊥Y2|T, in general [28]. (**b**) The DAG that is “implied” by the perspective of $I_{\cap }^{d}$. (**c**) A DAG that can generate the XOR distribution, but does not satisfy the dependencies implied by $T=Y_{1}\;\mathrm{xor}\;Y_{2}$. In fact, any DAG that is in the same Markov equivalence class as (**c**) can generate the XOR distribution (or any other joint distribution), but none satisfy the earlier dependencies, assuming faithfulness.

**Figure 2 entropy-26-00271-f002:**
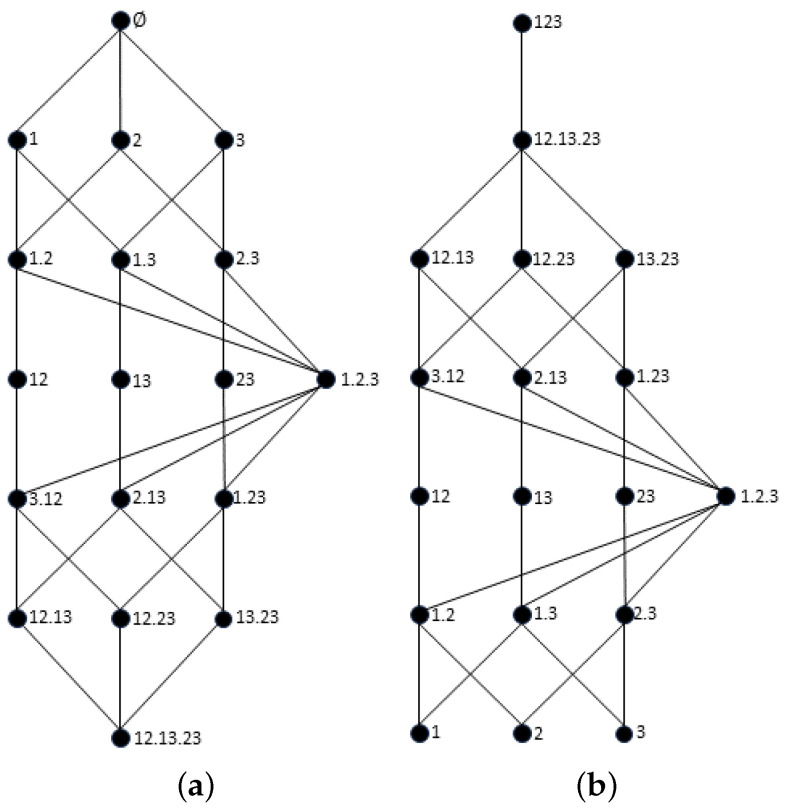
Trivariate distribution lattices and their respective ordering of sources. Left (**a**): synergy lattice [18]. Right (**b**): union information semi-lattice [36].

**Figure 3 entropy-26-00271-f003:**
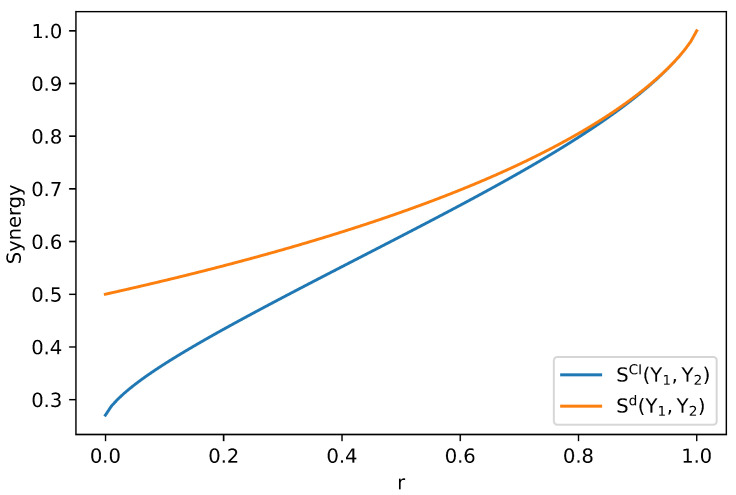
Computation of $S^{CI}$ and $S^{d}$ as functions of r=p(T=0|Y=(0,0)) for the distribution presented in Table 8. As we showed for this distribution, $S^{CI}$ is not a convex function of *r*.

**Table 2 entropy-26-00271-t002:** Tweaked COPY distribution, now without the outcome $\left(T,Y_{1},Y_{2}\right)=\left(\left(1,1\right),1,1\right)$.

T	$Y_{1}$	$Y_{2}$	$p(t,y_{1},y_{2})$
(0,0)	0	0	1/3
(0,1)	0	1	1/3
(1,0)	1	0	1/3

**Table 3 entropy-26-00271-t003:** Left: The *Adapted Reduced OR* distribution, where r∈[0,1]. Right: The corresponding distribution $q\left(t,y_{1},y_{2}\right)=p\left(t\right)p\left(y_{1}|t\right)p\left(y_{2}|t\right)$.

t	$y_{1}$	$y_{2}$	$p(t,y_{1},y_{2})$	t	$y_{1}$	$y_{2}$	$q(t,y_{1},y_{2})$
0	0	0	1/2	0	0	0	1/2
1	0	0	r/4	1	0	0	1/8
1	1	0	(1 − r)/4	1	1	0	1/8
1	0	1	(1 − r)/4	1	0	1	1/8
1	1	1	r/4	1	1	1	1/8

**Table 4 entropy-26-00271-t004:** Counter-example distribution for target monotonicity.

T	Z	$Y_{1}$	$Y_{2}$	$p(t,z,y_{1},y_{2})$
0	(0,0)	0	0	1/4
0	(0,1)	0	1	1/4
0	(1,0)	1	0	1/4
1	(1,1)	1	1	1/4

**Table 5 entropy-26-00271-t005:** Counter-example distribution for target monotonicity.

T	Z	$Y_{1}$	$Y_{2}$	$p(t,z,y_{1},y_{2})$
0	0	1	0	0.419
1	1	2	1	0.203
2	1	3	0	0.007
0	0	3	1	0.346
2	2	4	4	0.025

**Table 6 entropy-26-00271-t006:** $T_{1}=$ COPY, $T_{2}=$ XOR.

$T_{2}$	$T_{1}$	$Y_{1}$	$Y_{2}$	$p(t_{2},t_{1},y_{1},y_{2})$
0	0	0	0	1/4
1	1	0	1	1/4
1	2	1	0	1/4
0	3	1	1	1/4

**Table 7 entropy-26-00271-t007:** Adapted XOR distribution.

*T*	$Y_{1}$	$Y_{2}$	$p(t,y_{1},y_{2})$
0	0	0	r/4
1	0	0	(1−r)/4
1	1	0	1/4
1	0	1	1/4
0	1	1	1/4

**Table 8 entropy-26-00271-t008:** Adapted XOR distribution v2.

*T*	$Y_{1}$	$Y_{2}$	$p(t,y_{1},y_{2})$
0	0	0	r/10
1	0	0	(1−r)/10
1	1	0	4/10
1	0	1	4/10
0	1	1	1/10

**Table 9 entropy-26-00271-t009:** Application of the measures reviewed in Griffith and Koch [16] ($S^{\mathrm{WB}}$, $S^{\mathrm{WMS}}$, and $S^{\Delta I}$), $S^{\mathrm{SD}}$ introduced by Rosas et al. [38], $S^{d}$ introduced by Kolchinsky [22], and our measure of synergy $S^{\mathrm{CI}}$ to different distributions. The bottom four distributions are trivariate. We write DNF to mean that a specific computation did not finish within 10 min.

Example	$S^{\mathrm{WB}}$	$S^{\mathrm{WMS}}$	$S^{\Delta I}$	$S^{d}$	$S^{\mathrm{SD}}$	$S^{\mathrm{CI}}$
XOR	1	1	1	1	1	1
AND	0.5	0.189	0.104	0.5	0.311	0.270
COPY	1	0	0	0	1	0
RDNXOR	1	0	1	1	1	1
RDNUNQXOR	2	0	1	1	DNF	1
XORDUPLICATE	1	1	1	1	1	1
ANDDUPLICATE	0.5	−0.123	0.038	0.5	0.311	0.270
XORLOSES	0	0	0	0	0	0
XORMULTICOAL	1	1	1	1	DNF	1

## Data Availability

Data is contained within the article.

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
