# Peer review of "A Measure of Synergy Based on Union Information"

_entropy, 2024, doi:10.3390/e26030271_

Round 1

Reviewer 1 Report

Comments and Suggestions for Authors

The authors present a novel quantification of the partial information decomposition, put it in the context of several other measures, and also consider it in light of the myriad of suggested properties that such measures should have. Overall, I find this paper to be a strong contribution to the field, though have several minor suggestions to improve the manuscript, and one overall comment which relates to the field in which this paper is being publish, but is not strictly a commentary on the paper itself.

Firstly, I find the fact that this measure is derived by considering union information, as opposed to directly quantifying one of redundancy, unique, or synergy, to be refreshing and provides a fairly novel point of view to a field which, largely, has been treading water for the past few years.

The discussion that begins in section 1.1 and continues in 2 regarding implicit directionality assumed by measures adopting a channel view is nice to have explicitly called out. It can perhaps be strengthened by including the result found in `Unique Information and Secret Key Agreement` by James, Emenheiser, and Crutchfield where, among unique informations defined using secret key agreement rates, only the one-way secret key agreement rate from T to Y_i results in a viable PID, supporting the common approach of using channels from target to source rather than the other way around.

I find the formulation in equation 4 to be very pleasing and intuitive. It's a shame that it results in a potentially negative synergy value and you were forced to adopt the form in eq 6. Still, the measure of synergy here is intuitive and offers a new perspective on the PID. In fact, after seeing eq 4 I was immediately surprised that this measure had not been explored perviously, it feels very natural.

The counter examples provided for target monotonicity and Rosas' target data processing inequality are particularly clear and intuitive, and I believe are quite damning for those two proposed properties. The discussion rejecting Quax's zero synergy about a single variable is fine, but depends upon eq 7 and the synergy lattice which itself isn't necessarily widely adopted by the community and so the discussion comes across somewhat weak.

Overall, the quantitative comparisons are enlightening. The comparisons to I_dep need some updating though. For the AND distribution, I believe your calculation did not optimize well, when I compute S^dep I get 0.27043, and I am confident that for this distribution I_r and I_p are identical. In fact, the (informational) difference between the two distributions is the 3rd order connected information (See E. Schneidman, S. Still, M. J. Berry, and W. Bialek. Network information and connected correlations. Phys. Rev. Lett., 91(23):238701, 2003.), which is known to be 0 for the AND distribution.

Finally, a few comments regarding typesetting and typos:

- In tables 1, 3, 6, 7, and perhaps 8, can you convert the decimals to fractions?

- Reference 11 is missing an author (Joy A. Thomas)

Reviewer 2 Report

Comments and Suggestions for Authors

In their manuscript, Gomes AF and Figueiredo MAT introduce a novel measure for computing union information within the framework of Partial Information Decomposition (PID), aimed at quantifying synergy in both bivariate and multivariate systems. This study significantly contributes to the ongoing debate within the PID community regarding the accurate and rigorous measurement of PID components. Notably, this research emphasizes union information—a frequently overlooked component in PID—and explores various properties of both union information and synergy as previously discussed in the literature. A distinctive feature of their synergy definition relies on the fact that synergistic information is zero in systems where sources are conditionally independent given the target source. Furthermore, the authors plan to release their code for computing synergy, which I believe will be a valuable contribution to the PID community.

Despite the critical importance of providing a synergy measure that avoids implausible values, such as negative synergy, and ensures brief computational times, the manuscript currently falls short in clarity regarding several key points. Specifically, the manuscript lacks a clear mathematical derivation of redundant and unique information for bivariate and multivariate cases. It remains unclear whether the inclusion-exclusion principle is implicitly assumed for computing redundancy and unique information. If not, the methodology for calculating redundancy and unique information from the proposed approach needs to be clarified.

Additionally, the manuscript could be improved by more effectively comparing the proposed framework to recent studies and by building a stronger case for the necessity of a new synergy measure. It does not immediately become apparent why the proposed approach should be preferred over the recently proposed synergy measure by Kolchinsky (2022) in Entropy. At lines 202 and 203, the authors argue that the synergistic information in the tweaked COPY distribution is negative when computed using Kolchinsky’s measure, while their new measure does not encounter this issue. Is this the sole scenario where the new union information metric proposed in this paper is advantageous, or are there other instances? The Discussion section should clearly articulate the advantages of the newly proposed synergy measure over previous ones, supported by references.

Lastly, making the code available during the revision process would facilitate a more comprehensive assessment of this work's impact on a broader community that applies PID to empirical data.

Other minor concerns:

1)      It is not clear to me what does the variable "Q" represent in Eq. (2).

2)      In Section 2.2 the authors make a few examples of redundancy measures satisfying or not the Blackwell property. For the sake of completeness, more references could be added here. Examples include Ince 2017 Entropy, Makkeh et al. 2021 PRE, Finn and Lizier 2018 Entropy, etc,

3)      I would suggest the authors integrating a part of section 2.2, which starts with “Towards motivating a new PID,…” until the end of the section, with section 3.1:“Motivation and Bivariate Definition”, which I think it is a more appropriate section.

4)      The notation in line 213 is not clear to me, how is y_(-i) defined?

5)      In lines 236 and 242 the union information and synergy are defined (see Eq (3-4)), but not in their final version, which is a bit confusing, because their final versions are in lines 261 and 263 (see Eq (5-6)). To be more effective in presenting their new measure of synergy and union information, I would suggest revising the text to avoid multiple definitions.

6)      What is the difference between Y1,Y2 and (Y1,Y2) in Eq. (3)?

7)      In section 4.1: what is the motivation behind the need to extend the axioms of Williams and Beer to the union information?

8)      In section 4.2: why the two properties of duplicating or adding a predictor stand out compared to the other seven properties, which have been proposed by the same authors Griffith and Koch? I would suggest putting all properties together in a single list of properties.

9)      In section 4.2: I would not include the properties 2 and 3 (see lines 399-400) in the list, because they have been already mentioned in lines 366-367. Otherwise, the authors can make a note saying that those properties were considered above in section 4.1.

10)  Remove one "built" from line 436.

11)  In section 4.4: In lines 484 and 495 I would plainly write “union information” instead of UI, because UI might be confused with U_i for the unique information.

12)  I would put Theorem 2 immediately below the introduction of the Griffith and Koch properties in the end of the section 4.2 (after line 409).

13)  The whole section 5, which refers to previous measures of union information and synergy, seems to better fit earlier in the manuscript. Indeed, I would consider placing it in section 2, which refers to the background. I think that it would be more impactful to first present previous measures of union information and synergy in section 2 and then propose a new definition in section 3.

14)  In section 5: Is there any reason why the list the synergy measure does not include the MMI-PID of Barrett 2015 PRE? I would consider it as well in these examples.

15)  It would be very helpful if the paper had a clarification note in the Discussion section on how to adapt and use the proposed synergy in systems with continuous variables.

Reviewer 3 Report

Comments and Suggestions for Authors

Please see the attached document

Comments on the Quality of English Language

Please see the above attached document.

Round 2

Reviewer 1 Report

Comments and Suggestions for Authors

I am satisfied with this current revision. 

Author Response

Thank you.

Reviewer 2 Report

Comments and Suggestions for Authors

Review Report

In my opinion, the paper has been only marginally revised concerning the main issues I previously highlighted. Therefore, I believe that substantial revisions are still necessary. Based on authors’ reply concerning the use of the IEP, it is crucial that the revised paper explicitly states that the proposed measure allows for a complete decomposition exclusively in the bivariate case, and not in the multivariate scenario. This limitation is significant because Kolchinsky's 2022 work on entropy was able to offer a multivariate measure that could fully decompose the Partial Information Decomposition (PID) into union information, synergy, and redundancy. Consequently, the current proposal appears to lack novelty. Considering Kolchinsky’s work also provides an efficient algorithm for computing redundancy (but not synergy) in the multivariate case, can a compelling justification be provided for the introduction of a new code for synergy/union information? Specifically, is there a gap in the literature regarding a computational tool for analyzing synergy in both bivariate and multivariate contexts that this work seeks to fill?

Therefore, the revised paper must:

1)      Clearly state that the proposed measure cannot fully decompose PID in the multivariate case.

2)      Elucidate why a full PID decomposition cannot be achieved in the multivariate case with the proposed approach.

3)      Justify why the proposed measures of union information and synergy are conceptually and/or computationally superior to those provided by Kolchinsky's 2022 entropy.

If these issues are not sufficiently addressed, the proposed measure's novelty appears minimal to me, leading me to question its suitability for publication in Entropy.

Minor issues:

1)      Regarding the provided code, it should be specified that it relies on the Python "dit" package. Initially, I was under the impression that it was standalone software. However, upon reviewing the readme file, it is mentioned that one must install the “dit” package beforehand. I believe it is relevant for this information to be included in the main text, as it is pertinent for individuals interested in applying the code to their data. Moreover, a more detailed description in the Readme file should be included.

Response to comment 1: Thank you for pointing that out. We have clarified it in the  new version, right after equation (2).

The notation in Eq. (2) is still a bit confusing because the notation Y1,…Ym->T implies that there is a direction from the Y1,…,Ym (considered as the inputs) to the output T, while in the text it is written that T is the input and Q, and  Y1,…,Ym are the outputs.

Response to comment 7: We decided to propose the equivalent of the Williams-Beer axioms, but for measures of union information, because that had not been done before. In fact, these axioms follow the requirements of PID as constructed by parthood functions, and can be seen in the work of Gutknecht. We added a clarification for this and explicitly wrote the consistency equation yielded by the work of Gutknecht, in lines 367-371.

Why should we care about this? The authors should motivate the rationale behind the need for possible extensions of the Williams-Beer axioms for measures of union information

Response to comment 8: Because they were not formally defined in the work of Griffith and Koch, and we suggested an alteration to one of them.

This reply is not clear to me because in the paper the authors write that all properties are defined in Griffith and Koch 2014. See lines 307 and 406.

Response to comment 14: We did not include the MMI PID because it is a redundancy-based PID, and we compared our measures against other synergy measures.

If that is the case, then why is SWB included in table 9? SWB, like MMI-PID, is a redundancy based PID.

Reviewer 3 Report

Comments and Suggestions for Authors

Please see the attached document

Author Response

Thank you for your comments. We corrected the compilation error. Regarding the Finn and Lizier measure, it is known that it doesn't satisfy the monotonicity axiom from the Williams and Beer axiom. Like you said, it satisfies a pointwise monotonicity axiom, but that's not what we were talking about.

Round 3

Reviewer 2 Report

Comments and Suggestions for Authors

I thank the authors for their detailed replies to my previous questions. Based on their clear argumentation, I am now inclined to recommend the publication of this paper in Entropy.